# OPENREVIEWER: PREDICTING CONFERENCE DECISIONS WITH LLMS AND BEYOND

## ABSTRACT

The rapid growth of AI conference submissions has strained the peer-review system, motivating interest in AI-assisted review. Yet it remains unclear how reliably such systems approximate human judgment, which relies on domain expertise and nuanced reasoning. To address this challenge,, we introduce `OpenReviewer`, a model designed to directly predict conference acceptance decisions rather than generate full reviews. Using ICLR 2024–2025 data, we evaluate large language models (LLMs), vision–language models (VLMs), and interpretable statistical models. Results show that text-only LLMs with continual pre-training outperform multimodal counterparts, achieving up to 78.5% accuracy on balanced datasets (vs. 50% random baseline). White-box statistical models further provide interpretability through feature analysis, revealing that structural attributes (e.g., paper length, section balance, citation engagement) are consistently predictive. Beyond average accuracy, a confidence-stratified utility analysis shows that the top 10% most confident predictions reach 92.92% overall precision, enabling reliable triage of "obvious" accepts and rejects while exposing areas of uncertainty. Overall, our findings demonstrate both the promise and limitations of AI-involved peer review: current models can reduce workload and aid submission reviewing, but fall short of reliably replacing expert judgment.

## 1 INTRODUCTION

The peer-review process is becoming increasingly unsustainable as submissions to top-tier AI conferences continue to grow at an unprecedented pace, as shown in Figure 1[1]. This explosive growth places pressure on program committees and reviewers, leading to heavier workloads and concerns over the quality and consistency of reviews (Lawrence, 2022; Beygelzimer et al., 2023; Kim et al., 2025; Schaeffer et al., 2025). For authors, uncertainty around submission outcomes and suboptimal venue choices can negatively influence research trajectories and academic career development (e.g., timely PhD graduation) (Kousha & Thelwall, 2024; Yang, 2025).

Recent work has explored AI-assisted review generation as a potential solution, where models take papers as inputs to generate reviews. (Sukpanichnant et al., 2024; Ye et al., 2024; Shin et al., 2025). To our knowledge, no existing work uses LLMs to predict acceptance directly from the paper content itself. Reliable acceptance prediction could guide authors in developing submission strategies, while helping committees triage obviously good/low-quality papers and allocate human review resources more effectively. Therefore, we propose

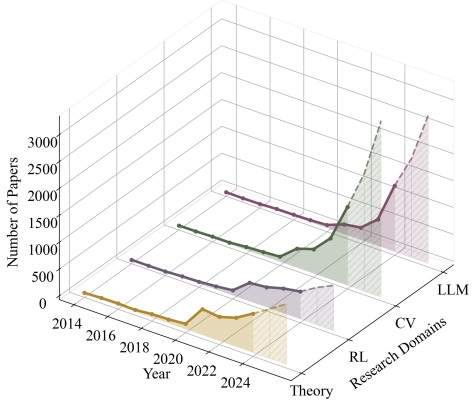

Figure 1: Number of papers accepted by NeurIPS from 2015 to 2024 across four major research domains, with the dashed line indicating the predicted trend.

---

[1]Our data sources include official conference announcements and the Paper Copilot platform (https://papercopilot.com/).

`OpenReviewer`, a LLM-based model that predicts the acceptance of papers submitted to AI conferences.

`OpenReviewer` is developed using conference submissions and corresponding acceptance information collected from the OpenReview platform[2]. In particular, our dataset consists of submission records from the International Conference on Learning Representations (`ICLR`)), chosen for its broad coverage of AI topics and the openness of its submission and decision records. The training dataset includes three components of papers: (1) *textual content*, consisting primarily of anonymized manuscript text; (2) *visual information*, such as system figures and charts; and (3) interpretable, manually engineered *statistical features*.

We adopt prompt-based fine-tuning (Shi & Lipani, 2023) to help LLMs understand the given task, combined with a decoupled label loss (Tam et al., 2021) to encourage the use of vocabulary tokens (e.g. `Yes, No`) as labels during training. We explore three approaches for this task: text-only large language models, vision–language models, and white-box statistical classifiers. Among text-only models, continued pretraining (CPT) on unlabeled corpora prior to fine-tuning yields the best result, achieving **78.5%** accuracy on a label-balanced dataset (50% random baseline). For VLMs models, unsurprisingly, incorporating image inputs consistently outperforms text-only inputs. We also provide qualitative analyses highlighting cases where images help and where they mislead. In addition, we conduct a white-box analysis of statistical features by extracting 29 heuristic quantitative features across eight categories. Using only these features, a Random Forest classifier (Breiman, 2001) attains a surprisingly strong 74.2% accuracy, surpassing VLMs. Finally, a model confidence-stratified analysis for `OpenReviewer` shows that within the top 10% confidence slice, covering up to 53.06% of predictions, LLMs achieve 93.09% precision on the Accept class, with a comparable trend observed for the Reject class. This enables reliable triage of clear accepts and rejects while routing uncertain cases for human review. Overall, these findings indicate that AI can support peer review by reducing the workload on straightforward submissions, while human experts remain essential for more more nuanced judgments.

## 2 RELATED WORK

**Peer Review Analysis.**   The peer review process, particularly in rapidly evolving fields like AI, is facing a sustainability crisis with reviewer overload and declining quality (Chen et al., 2025; Kim et al., 2025). While LLMs have been explored to automate or assist reviewing, their readiness lacks validation. Large-scale experiments show LLMs can distinguish paper quality but exhibit significant biases (Pataranutaporn et al., 2025), with researchers warning against premature deployment due to risks in factual accuracy and logical reasoning (Ye et al., 2024). New evaluation methods identify "blind spots" in LLM reviews, revealing that they often miss crucial methodological flaws such as experimental design issues, statistical significance problems, and logical inconsistencies in argumentation (Shin et al., 2025). Improvement efforts include structured argumentative review frameworks (Sukpanichnant et al., 2024) and graph reasoning systems over reviewer-author debates (Taechoyotin & Acuna, 2025). Additionally, AI-assisted reviews create an "AI review lottery," inflating scores and masking weaknesses (Latona et al., 2024). These challenges prompt calls for systemic reform, including transparent processes and reviewer rewards (Yang, 2025; Ye et al., 2024), dedicated critique tracks (Schaeffer et al., 2025), and lessons from platforms like OpenReview (Wang et al., 2023).

**Paper Quality Modeling.**   Recent advancements in LLMs have spurred significant research into computationally modeling the quality of scholarly papers in the form of evaluation, revision, and generation. Research focuses on automated assessment using domain-aware retrieval and latent reasoning (Zheng et al., 2025), verifiable claim extraction (Song et al., 2024), and retraction prediction for scientific integrity (Yang & Jia, 2025). Beyond evaluation, quality models support paper improvement through human-AI collaborative revision frameworks (Fragiadakis et al., 2024; Dong et al., 2022) and fully automated generation systems like ARISE, which uses explicit quality rubrics (Schneider, 2025).

---

[2]https://openreview.net/

**LLM-based Document Classification.** LLMs shift document classification from traditional fine-tuning to prompt-based, few-shot learning. This involves reformulating classification tasks as cloze questions, enabling strong performance with minimal labeled data (Schick & Schütze, 2021). Recent studies further demonstrate that continued pretraining can significantly enhance prompt-tuning effectiveness, making it an even more powerful learning approach (Chen et al., 2022). Despite these advances, LLM-based classification faces several challenges. Raw LLM outputs often suffer from miscalibration issues, necessitating the development of context-aware calibration techniques (Zhao et al., 2021). Additionally, the direct application of LLMs as zero-shot or few-shot classifiers shows promise but remains task-dependent, requiring careful model selection for specialized domains such as classifying scientific revision intents (Ruan et al., 2024). To address these limitations, researchers have developed hybrid and advanced approaches. Hybrid models like DeepCCP successfully integrate semantic understanding with citation network structure to achieve more accurate classification (Zhao & Feng, 2022). Furthermore, advanced approaches explore classification through generation tasks, including benchmarking LLMs on writing paper sections (Garg et al., 2025) and developing multi-agent frameworks for paper reproduction (Miao et al., 2025). These developments highlight the evolution towards deeper, context-aware reasoning.

## 3 OPENREVIEWER FOR PREDICTING ACCEPTANCE

We formulate paper acceptance prediction as a binary classification problem. State-of-the-art LLMs and VLMs are inherently generative, making them not directly applicable to traditional classification tasks. To use the capabilities of these powerful pre-trained generative models without training a new classification head from scratch[3], we adopt a prompt-based fine-tuning strategy Ruan et al. (2024); Schick & Schütze (2021); Shi & Lipani (2023). Specifically, we design an instructive prompt template $\mathcal{T}$ that presents the paper's features within a natural-language query and guides the model to generate a decision token corresponding to one of the two target classes: *accept* or *reject*. The template example is given in App. D.

### 3.1 CONTINUAL PRE-TRAINING

Continual pre-training (CPT) extends the training of large generative models on additional unlabeled corpora to improve their adaptability to new domains and evolving data distributions (Gururangan et al., 2020; Chen et al., 2023). It is widely adopted in industry-scale generative systems, where models are periodically updated with fresh data to sustain relevance and maintain competitive performance (Gururangan et al., 2020; Chen et al., 2023; Ke et al., 2023; Elhady et al., 2025). The training objective typically follows next-token prediction, formalized as

$$\mathcal{L}_{\text{CPT}} = -\sum_{t=1}^{T} \log P_\theta(x_t \mid x_{<t}), \tag{1}$$

which maximizes the likelihood of generating each token $x_t$ given its preceding context $x_{<t}$ and model parameters $\theta$. In this paper, we also explore continual pre-training to adapt general-purpose base models to the academic peer-review scenario before fine-tuning on the classification task. We present the effectiveness of CPT in Section 4.3, with further training details provided in the App. E. Unless otherwise specified, CPT is used as the default post-training strategy for our textual models before fine-tuning.

### 3.2 INPUT SETTINGS

Given a paper input instance $x$ and prompt template $\mathcal{T}(x)$, the model defines a conditional probability over the label verbalizer (Tam et al., 2021). We consider two input configurations for $\mathcal{T}(x)$: *text-only* and *text-image multimodal*. Text-only inputs are anonymized main-body texts from the paper manuscripts. The multimodal setting extends the text-only configuration by additionally incorporating visual features extracted from figures in the paper. Details of the PDF preprocessing and figure extraction procedure are provided in App. C. Formally,

$$\mathcal{T}(x) = \phi\big(x^{(\text{text})} \ \oplus \ x^{(\text{figure})}\big) \tag{2}$$

---

[3]Our initial experiments with training a classification head on top of a pre-trained LLMs resulted in lower accuracy and slower convergence compared to prompt-based generation.

where $\phi$ is modality-specific encoding determined by the multimodal model, and $x^{\text{(figure)}}$ is an optional input. $\mathcal{T}(x)$ packs all modalities into a single token sequence consumable by the model.

We then append a designated decision slot and generate only at this position, defining $\mathcal{T}^{\text{dec}} = \mathcal{T}(x) \oplus [\text{label\_mask}]$. *Verbalizer* $\mathcal{V}$ maps each candidate label token $v$ to a class; this allows many-to-one mappings (e.g., $\{yes, accept\} = 1; \{reject, no\} = 0$).

### 3.3 MODEL TRAINING OBJECTIVE

Given an input $\mathcal{T}^{\text{dec}}$ and its corresponding ground-truth label $y^*$, we apply supervision *only* at the decision position, masking all other positions. Following ADAPET (Tam et al., 2021), we define the Vocabulary Decoupled Label Loss (VDLL). Let $z_\theta(t \mid \mathcal{T}^{\text{dec}})$ denote candidate labels logits at the decision slot $t$, we then define the (3) *restricted softmax* and (4) training objective as:

$$\tilde{p}_\theta(t \mid \mathcal{T}^{\text{dec}}) = \frac{\exp\big(z_\theta(t \mid \mathcal{T}^{\text{dec}})\big)}{\sum_{a \in \mathcal{V}} \exp\big(z_\theta(a \mid \mathcal{T}^{\text{dec}})\big)} \tag{3}$$

$$\mathcal{L}_{\text{VDLL}}(\theta) = -\log \sum_{t \in \mathcal{V}_{y^*}} \tilde{p}_\theta\big(t \mid \mathcal{T}^{\text{dec}}\big), \tag{4}$$

### 3.4 INFERENCE MECHANISM

At inference time, we determine the predicted class by comparing the *logit-based* scores of all verbalizer candidates at the decision slot. Let $z_\theta(t \mid \mathcal{T}^{\text{dec}})$ denote the pre-softmax logit assigned by the model to token $t$ at the decision position. We first obtain the token IDs of all verbalizer candidates $\mathcal{V}$. For each class $y$, the score is defined as the maximum logit among its associated verbalizer tokens. The final prediction $\hat{y}$ is then obtained by selecting the class with a higher score, for example predicting `Accept` if $\text{score}(\text{yes}) > \text{score}(\text{no})$ and vice versa:

$$\hat{y} = \arg \max_{y \in \mathcal{Y}} \text{score}_\theta(y \mid \mathcal{T}^{\text{dec}}) \tag{5}$$

We report a binary decision $b(\hat{y}) \in \{0, 1\}$.

## 4 EXPERIMENTS

### 4.1 DATA COLLECTION AND PRE-PROCESSING

We collect all ICLR 2025 and 2024 submissions and their corresponding final decisions (*accepted* or *rejected*) via the `OpenReview API-V2`. The papers were further partitioned into four main subfields based on title keywords: Large Language Models (LLM), Computer Vision (CV), Reinforcement Learning (RL), and Theoretical (Theory). Papers that do not fall into these categories are left for future discussion. We build two datasets: the ICLR 2025 dataset, which is naturally imbalanced with a 34/66 accepted-to-rejected split and balanced domain-specific sets from ICLR 2024 and 2025 with a 50/50 split. Table 6 summarizes the differences. More implementation details are explained in App. C.

### 4.2 MODELS AND INPUTS

We include two categories of models: **text-only** LLMs and **vision-language** models. For the text-only LLMs, we select the `Qwen-3` family at `Qwen3-4B` and `Qwen3-8B` parameter scales (Yang et al., 2025). For VLMs, we include `Qwen2.5-VL-3B-Instruct` (Bai et al., 2025) and `Gemma-3-4b-it` (Team et al., 2025). Both of these multimodal models are instruction-tuned variants. We take the vanilla non-fine-tuned version of each model in a zero-shot setting as the baseline. After collecting and preprocessing the papers along with their acceptance outcomes, we fine-tune and evaluate the two categories of selected models using the following inputs.

**Text-only LLMs:** We first anonymize each paper by removing all information that could reveal author identity or acceptance status, including author names, affiliations, email addresses, URLs, and header or footer text. Beyond these removals, the input consists of the full manuscript body text and mathematical formulas, but excludes tables and figure captions.

Table 1: Performance (%) of LLMs across four domains and the overall aggregation (ALL) on the balanced dataset. All models use Qwen3 as the backbone. We compare fine-tuning with CPT against the original checkpoints (Orig) at the 4B and 8B parameter scales.

| SUB-DOMAIN | ALL | | | | LLM | | | | CV | | | | RL | | | | THEORY | | | |
|---|---|---|---|---|---|---|---|---|---|---|---|---|---|---|---|---|---|---|---|---|
| | Acc | Mac-P | Mac-R | F1 | Acc | Mac-P | Mac-R | F1 | Acc | Mac-P | Mac-R | F1 | Acc | Mac-P | Mac-R | F1 | Acc | Mac-P | Mac-R | F1 |
| CPT 4B* | 48.7 | 23.8 | 51.0 | 34.7 | 54.2 | 26.9 | 52.4 | 36.5 | 51.0 | 26.6 | 49.5 | 33.7 | 50.6 | 22.2 | 47.4 | 32.6 | 46.1 | 24.3 | 48.6 | 30.2 |
| CPT 4B | 76.4 | 76.3 | 76.4 | 76.4 | 70.2 | 70.1 | 70.2 | 70.1 | 70.2 | 75.3 | 70.4 | 68.7 | 57.4 | 62.5 | 57.3 | 52.4 | 55.9 | 61.5 | 55.5 | 49.1 |
| CPT 8B | 78.5 | 78.5 | 78.3 | 78.5 | 70.0 | 70.1 | 70.0 | 70.1 | 73.9 | 74.1 | 74.0 | 73.9 | 67.5 | 68.0 | 67.6 | 67.3 | 53.4 | 53.6 | 53.2 | 51.0 |
| Orig 4B* | 51.9 | 24.5 | 52.1 | 33.5 | 54.4 | 28.1 | 51.6 | 38.0 | 51.1 | 27.2 | 49.6 | 37.5 | 51.5 | 21.6 | 49.6 | 31.7 | 47.4 | 22.3 | 47.6 | 31.5 |
| Orig 4B | 67.3 | 71.4 | 68.2 | 66.3 | 68.8 | 69.1 | 68.8 | 68.6 | 71.0 | 71.0 | 71.1 | 71.0 | 59.3 | 59.7 | 59.3 | 58.9 | 57.2 | 59.5 | 56.9 | 53.9 |
| Orig 8B | 69.0 | 69.0 | 69.0 | 69.0 | 69.7 | 69.9 | 69.8 | 69.7 | 72.5 | 73.2 | 72.6 | 72.4 | 62.6 | 64.3 | 62.6 | 61.5 | 55.4 | 55.9 | 55.2 | 54.0 |

* Baseline models. Random guess baseline accuracy is 50%.

Table 2: Performance (%) of VLMs across four domains on the balanced dataset. Mac-P and Mac-R denote Macro Precision and Macro Recall, respectively.

| SUB-DOMAIN | ALL | | | | LLM | | | | CV | | | | RL | | | | THEORY | | | |
|---|---|---|---|---|---|---|---|---|---|---|---|---|---|---|---|---|---|---|---|---|
| | Acc | Mac-P | Mac-R | F1 | Acc | Mac-P | Mac-R | F1 | Acc | Mac-P | Mac-R | F1 | Acc | Mac-P | Mac-R | F1 | Acc | Mac-P | Mac-R | F1 |
| Qwen2.5-VL txt&img* | 46.0 | 23.0 | 50.0 | 31.5 | 54.6 | 27.1 | 50.0 | 35.3 | 51.6 | 25.8 | 50.0 | 34.0 | 50.2 | 25.1 | 50.0 | 33.4 | 47.6 | 23.8 | 50.0 | 32.3 |
| Qwen2.5-VL txt* | 48.0 | 24.0 | 50.0 | 32.4 | 54.6 | 27.3 | 50.0 | 35.3 | 51.6 | 25.8 | 50.0 | 34.0 | 50.2 | 25.1 | 50.0 | 33.4 | 47.6 | 23.8 | 50.0 | 32.3 |
| Qwen2.5-VL txt&img | 68.2 | 68.6 | 67.8 | 67.7 | 74.2 | 75.5 | 74.5 | 74.1 | 70.0 | 70.1 | 69.8 | 69.8 | 65.7 | 66.2 | 65.8 | 65.5 | 61.5 | 62.6 | 60.5 | 59.3 |
| Qwen2.5-VL txt | 64.4 | 69.2 | 65.3 | 62.7 | 69.9 | 70.2 | 70.3 | 69.9 | 69.0 | 69.1 | 68.8 | 68.8 | 60.6 | 62.5 | 60.5 | 58.8 | 61.5 | 61.4 | 61.1 | 61.1 |
| Gemma-3 txt&img* | 34.4 | 17.2 | 50.0 | 25.6 | 50.6 | 53.1 | 50.0 | 33.9 | 50.4 | 58.5 | 50.0 | 33.6 | 50.0 | 25.0 | 50.0 | 33.3 | 49.6 | 41.5 | 49.9 | 33.3 |
| Gemma-3 txt* | 34.4 | 17.2 | 50.0 | 25.6 | 50.0 | 35.0 | 50.0 | 33.4 | 50.0 | 50.0 | 50.0 | 33.5 | 50.0 | 25.0 | 49.9 | 33.3 | 50.0 | 41.7 | 50.0 | 33.5 |
| Gemma-3 txt&img | 61.9 | 58.0 | 53.2 | 44.3 | 61.9 | 60.5 | 60.3 | 60.3 | 56.5 | 56.4 | 56.3 | 56.2 | 55.7 | 57.5 | 55.6 | 52.7 | 55.9 | 55.9 | 55.9 | 55.8 |
| Gemma-3 txt | 71.2 | 67.7 | 66.0 | 66.5 | 57.5 | 59.3 | 57.3 | 54.8 | 58.4 | 59.0 | 58.7 | 58.2 | 58.4 | 61.2 | 58.5 | 55.8 | 59.0 | 58.5 | 57.3 | 56.6 |

* Baseline models. Random guess baseline accuracy is 50%.

**Multimodal Models:** For VLMs, the input consists of only only the *Abstract* and *Introduction* text, together with the first two figures from each paper. To disentangle the contributions of textual and visual information in VLMs, we consider two input configurations: **text+image** and **text-only**.

## 4.3 RESULTS

We evaluate prediction performance using Accuracy, Macro-Precision (Mac-P), Macro-Recall (Mac-R), and F1. Mac-P and Mac-R average class-wise precision and recall, while F1 is the harmonic mean of precision and recall. As shown in Table 1 and 2, text-only unimodal models generally outperform multimodal text-image models of comparable size. For example, within the Qwen family, Qwen3-4B achieves 76.4% accuracy, surpassing multimodal Qwen2.5-VL-3B-Instruct at 68.2% and also shows consistently higher Mac-R, Mac-P, and F1.

**Text-only models** We evaluate two training strategies: (i) prompt-based fine-tuning on the original models, and (ii) CPT followed by prompt-based fine-tuning. As shown in Table 1, CPT yields clear improvements for downstream classification. On the aggregated ALL domain, CPT consistently outperforms fine-tuning from the original checkpoints (Orig) at the same parameter scale, **improving accuracy from 67.3% to 76.4% at 4B and from 69.0% to 78.5% at 8B**. Moreover, CPT is more effective at larger scales. For instance, CPT yields a 9.1% improvement at 4B while a 9.5% improvement at 8B on the all domain, with consistent increases in LLM, CV, and RL at the 8B scale.

**Vision–Language models** We evaluate various VL models, i.e., Qwen2.5-VL-3B-Instruct and Gemma-3-4B-it. As shown in Table 2, **Qwen2.5-VL outperforms Gemma-3**, achieving 68.2% versus 61.9% with text–image input, and consistently higher accuracy across all four sub-domains. Second, for Qwen2.5-VL, incorporating text–image input consistently improves performance over text-only input. For example, in the LLM domain it achieves 74.2% compared to 69.9% with text-only, and this trend holds across the other three subdomains as well as the aggregated all domain. More results on the imbalanced dataset in-domain result in-domain result are provided in App. F.1

Figure 2: Examples of vision–language model predictions on previous submission. Case Group 1: text alone leads to incorrect predictions, while the image provides complementary cues that correct the outcome. Case Group 2: text alone yields the correct answer, but adding the image introduces misleading signals and causes errors.

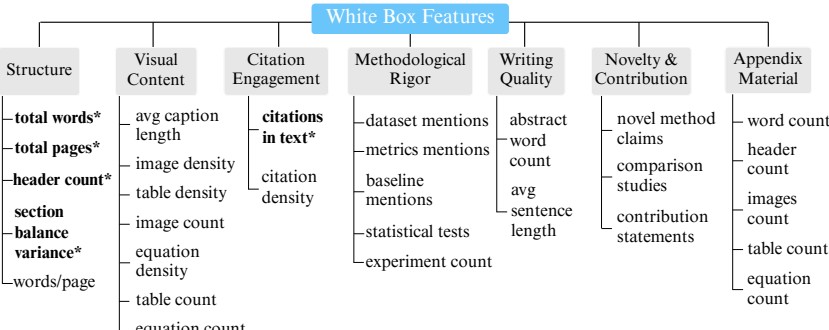

Figure 3: White box features organized by category and ranked by importance within each group. Features marked with asterisks (*) represent the top 5 most important features from the Random Forest model

## 4.4 QUALITATIVE ANALYSIS FOR VL-MODELS

To better understand the role of visual inputs, we qualitatively analyze two outcomes: *Image Helps*, where the model fails with text but succeeds with text–image inputs, and *Image Misleads*, where the addition of images reduces accuracy. Figure 2 illustrates these patterns on prior submissions (Xiong et al., 2025; Lin et al., 2024; Malagón et al.; Qian et al.; Wang et al., 2025; Fort). *Image Helps* (first row) show that schematic figures conveying high-level methodology or motivation, such as pipelines or dataset overviews, help predict the acceptance. In contrast, *Image Misleads* cases often involve detailed result visualizations that are difficult to interpret from figures alone. Additional examples are in App. F.2. We further evaluate models using images as the sole input modality with detail provided in App. F.3.

## 4.5 STATISTICAL FEATURE ANALYSIS

We train white-box statistical models on manually engineered features to provide an alternative performance baseline and interpretable insights into the structural characteristics that distinguish accepted papers from rejected (Wang et al., 2023).

**Models and Features** We extract 29 quantitative features from each submission PDF across seven categories, as illustrated in Figure 3. A comprehensive list of all features can be found in App. H. These features are then used to train four supervised classifiers, namely Random Forest (Breiman, 2001), Support Vector Machine (Schölkopf et al., 1999), Logistic Regression (Hosmer Jr et al., 2013), and Gradient Boosting (Friedman, 2002).

| Domain | Imbalanced Dataset | | | | | Balanced Dataset | | | | | Out-of-Distribution Test | | | | |
|---|---|---|---|---|---|---|---|---|---|---|---|---|---|---|---|
| | Size | Model | Acc | F1 | AUC | Size | Model | Acc | F1 | AUC | Size | Model | Acc | F1 | AUC |
| LLM | 3,716 | SVM | 70.6 | 34.3 | 72.2 | 3,238 | RF | 66.3 | 67.8 | 71.5 | 2,121 | GB | 49.5 | 4.6 | 57.4 |
| CV | 2,776 | RF | 71.5 | 49.0 | 72.2 | 3,520 | GB | 68.3 | 69.6 | 73.3 | 2,230 | LR | 51.4 | 2.8 | 55.8 |
| RL | 1,251 | LR | 70.1 | 44.4 | 72.1 | 1,526 | GB | 65.7 | 67.7 | 70.5 | 1,008 | GB | 51.5 | 1.1 | 58.3 |
| Theory | 1,735 | SVM | 68.5 | 36.9 | 70.2 | 1,974 | SVM | 63.0 | 64.7 | 71.2 | 1,228 | GB | 49.8 | 2.9 | 50.8 |
| Combined | 9,478 | RF | 77.3 | 60.9 | 83.0 | 10,258 | RF | 74.2 | 74.9 | 83.1 | 6,587 | GB | 53.1 | 2.2 | 61.8 |

GB = Gradient Boosting, RF = Random Forest, LR = Logistic Regression, SVM = Support Vector Machine

Table 3: Performance of statistical models on (i) the imbalanced ICLR 2025 dataset, (ii) balanced domain-specific datasets, and (iii) the Out-of-Distribution Test: models trained on the imbalanced ICLR 2025 data and evaluated on the balanced 50/50 test set.

**Classification Performance** Table 3 reveals distinct performance patterns across all dataset configurations. Random Forest emerges as the best-performing white-box model across both imbalanced and balanced datasets, achieving 77.3% accuracy with an F1-score of 60.9 on imbalanced data, and 74.2% accuracy with a substantially improved F1-score of 74.9 on balanced data. The out-of-domain study demonstrates that *models trained on imbalanced data but evaluated on balanced datasets suffer significant performance degradation*, with Random Forest achieving only 53.1% accuracy and immensely low F1-scores across all models, as the models classified nearly all papers as rejected due to their bias toward the majority class learned from the rejection-heavy imbalanced training data.

The balanced dataset yields on average slightly lower accuracy but significantly higher F1-scores compared to imbalanced, despite having less training data, indicating that *class balance is more critical than dataset size for effective predicting minority research domain*. Across both balanced and imbalanced configurations, combined domain models consistently achieve the best performance compared to individual domains, demonstrating that *cross-domain feature interactions enhance predictive capability*. However, all white-box model results remain significantly below those achieved by fine-tuned LLMs and VLMs, showing the limitations of traditional machine learning approaches in capturing the semantic complexity inherent in peer review decisions.

(a) Imbalanced Dataset

| Rank | Feature | Imp. | Cat. |
|---|---|---|---|
| 1 | total words | 0.0739 | Struct. |
| 2 | header count | 0.0587 | Struct. |
| 3 | total pages | 0.0575 | Struct. |
| 4 | section balance variance | 0.0492 | Struct. |
| 5 | avg caption length | 0.0465 | Visual |

(b) Balanced Dataset

| Rank | Feature | Imp. | Cat. |
|---|---|---|---|
| 1 | total words | 0.0792 | Struct. |
| 2 | total pages | 0.0658 | Struct. |
| 3 | header count | 0.0569 | Struct. |
| 4 | section balance variance | 0.0474 | Struct. |
| 5 | citations in text | 0.0448 | Citation |

Table 4: Top five most discriminative features for paper acceptance prediction from Random Forest analysis across both dataset configurations.

**Feature Importance Analysis** Random Forest feature importance analysis reveals that structural characteristics dominate acceptance prediction across both dataset configurations, as measured by Gini impurity–based importance scores (Nembrini et al., 2018). As shown in Table 4, the same core structural features consistently appear in the top five most discriminative features across both imbalanced and balanced datasets, suggesting that **paper acceptance favors structure quality rather than domain-specific content**.

Examining the feature rankings reveals several patterns. Content length indicators (`total words`, `total pages`) consistently dominate both configurations, with `total words` ranking first in both cases but showing increased importance (0.079 vs 0.073) in the balanced dataset. Organizational structure features (`header count`, `section balance variance`) maintain high importance across configurations. Most notably, `citations in text` replaces `avg caption length` in the balanced dataset's top five, suggesting that scholarly engagement becomes more discriminative when class imbalance is addressed.

These patterns indicate that accepted papers consistently tend to be more comprehensive (evidenced by length-based features), better organized (reflected in structural balance metrics), and demonstrate

stronger scholarly engagement (particularly evident in balanced datasets where citation patterns emerge as discriminative). However, the modest importance scores (all $< 0.08$) across both configurations indicate that **no single structural characteristic serves as a strong predictor**, explaining why semantic understanding via LLMs significantly outperforms purely structural approaches.

## 5 UTILITY ANALYSIS FOR RECOGNIZING "OBVIOUS" PAPERS

In Section 4, we set a default acceptance threshold using $\text{score}(\texttt{yes}) > \text{score}(\texttt{no})$, though this can be adjusted in practical peer-review workflows. In practice, if the model can confidently triage "clearly good" and "clearly bad" submissions with minimal errors, it can both reduce reviewer workload and discourage authors from making redundant submission attempts. This section provides a *confidence-based utility analysis* to accommodate this need.

### 5.1 CONFIDENCE-BASED STRATIFICATION

**Decision confidence.** At the designated decision slot (cf. §3), let $l_{\text{yes}}$ and $l_{\text{no}}$ be the pre-softmax logits for the tokens associated with the labels ACCEPT and REJECT, respectively. We define $p$ as the softmax-normalized probability assigned to a class, accept or reject, when considering only these two logits. Then we define a scalar *confidence* $c$ with $c \approx 0$ indicates indecision ($\approx 0.5/0.5$) and $c \approx 1$ indicates near-certainty. Formally,

$$c = \left| p_{\text{yes}} - p_{\text{no}} \right| = \left| 2p_{\text{yes}} - 1 \right| \in [0, 1] \tag{6}$$

**The coverage metric.** Next, we define *coverage* as the fraction of a class's falling within a given confidence bin. Predictions are partitioned into disjoint bins $B_k$ (e.g., $[0.0, 0.1), \ldots, [0.9, 1.0]$). For a set $\mathcal{S}$ of examples (restricted to a predicted class), the *coverage* of bin $B_k$ is:

$$\text{Cov}(B_k; \mathcal{S}) = \frac{1}{|\mathcal{S}|} \sum_{i \in \mathcal{S}} 1\{c_i \in B_k\} = \frac{\left| \{ i \in \mathcal{S} : c_i \in B_k \} \right|}{|\mathcal{S}|}. \tag{7}$$

$c_i \in [0, 1]$ is confidence value of $i$. Using these definition together with precision per class, we then examine how reliability scales with the model's self-reported certainty.

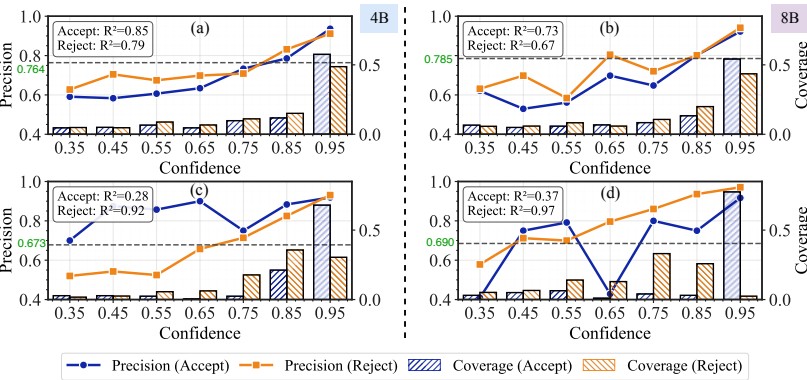

Figure 4: Coverage and precision across confidence bins for ACCEPT and REJECT predictions, shown for 4B and 8B CPT models. Each panel reports the linear coefficient $R^2$ of a least-squares fit of precision vs. confidence. Panels (a) and (b) correspond to models trained on the balanced dataset, while panels (c) and (d) correspond to models trained on the imbalanced.

### 5.2 STRATIFIED RESULTS AND OBSERVATIONS

We analyze four CPT models: `Qwen3-4B` and `Qwen3-8B` with each trained on the balanced and imbalanced datasets, and summarize their behavior in Fig. 4, which plots coverage and precision for both predicted classes across confidence bins. Overall, we observe that high-confidence regions ($c \geq 0.9$) achieve high precision with substantial coverage, while class imbalance reduces the coverage of confident rejects.

**Confidence Concentration and Coverage–Confidence Patterns**   Across all models, an average of *81.3%* of predictions fall within the high-confidence range $c \in [0.8, 1.0]$. In the most confident interval $c \in [0.9, 1.0]$ ($c = 0.95$ in the table), both ACCEPT and REJECT achieve precision above *91%*. This indicates the presence of substantial "obvious tails" that can be triaged with minimal error: **when models are highly confident, they are usually correct.** For ACCEPT, coverage increases *monotonically* with confidence: it always exceeds *50%* and reaches *75.2%* for the Qwen3-8B model on the imbalanced dataset (Figure 4d), suggesting that most acceptance predictions are made with high certainty. In contrast, although REJECT precision improves as $c$ increases, its coverage is not consistently monotonic under imbalanced training, reflecting the relative scarcity of confidently identified rejections.

We further assess how precision scales with confidence by fitting a least-squares regression separately for ACCEPT and REJECT. The coefficient of determination ($R^2$) (Piepho, 2019), reported in the figures, characterizes the degree of linearity in this relationship. The results of $R^2$ indicate that for the minority class, models trained on imbalanced data exhibit markedly poorer certainty than their counterparts trained on balanced data. More details are provided in App. I and J.

### 5.3   OPENREVIEWER HELPS IDENTIFY "OBVIOUS" GOOD/BAD PAPERS

In this section, we examine whether OpenReviewer can reliably identify papers that are clear accepts or clear rejects. To this end, we focus on predictions where the model is extremely confident ($c \in [0.9, 1.0]$) and analyze the corresponding error rates using the case of Qwen3-4B model trained on the balanced dataset. First, we rank them by their confidence scores $c$ and take the top-$K\%$ mass within this band with $K \in \{1, 3, 5, 7, 9\}$, i.e., 2% step increases. For each slice we report per-class *error* ($= 1 - \text{precision}$) and *coverage*.

| Top-mass slice | ACCEPT | | REJECT | |
|---|---|---|---|---|
| | Error ↓ | Coverage ↑ | Error ↓ | Coverage ↑ |
| Top 1.0% | 2.18 | 12.74 | 3.07 | 11.36 |
| Top 3.0% | 3.21 | 28.91 | 3.94 | 26.58 |
| Top 5.0% | 4.12 | 36.84 | 4.83 | 33.71 |
| Top 7.0% | 4.89 | 40.41 | 5.51 | 39.18 |
| Top 9.0% | 6.03 | 45.02 | 6.06 | 41.12 |
| All (10%) | 6.91 | 53.06 | 7.24 | 47.34 |

Table 5: Performance of high-confidence predictions ($c \in [0.9, 1.0]$): error rates and coverage for progressively larger confidence slices. Error rate (%) lower is better ↓; coverage (%) shows the fraction of each class captured in the slice (higher is better ↑).

Table 5 reveals encouraging results for workload reduction. When we consider only the top 1% most-confident predictions, the model covers 12.74% of all accept decisions with just 2.18% error, and 11.36% of all reject decisions with 3.07% error. *In practical terms, if the model makes 500 accept predictions, the 64 most-confident ones would contain fewer than two mistakes.*

As we expand to include more confident predictions, we naturally trade some accuracy for greater coverage. The top 9% slice covers nearly half of all decisions, 45.02% of accepts and 41.12% of rejects, while maintaining reasonably low error rates of 6.03% and 6.06% respectively, illustrating the expected precision-coverage trade-off.

These results suggest that a confidence-based triage system could substantially reduce reviewer workload. By automatically handling the most obvious cases where the model is highly confident, conferences could focus human reviewer effort on the more nuanced submissions where expert judgment is most valuable.

## 6   CHALLENGES AND FUTURE WORK

This paper presents the first work using LLM to predict AI paper acceptance. Our work opens several promising directions for AI-assisted reviewing, including (i) assessing fairness across subfields, (ii) monitoring evolving conference standards, (iii) effectively integrating human-in-the-loop review pipelines, and (iv) exploring bias detection to ensure equitable outcomes.

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

## A    REPRODUCIBILITY STATEMENT

We take several steps to enable full replication of our results.

**Data.**    We use ICLR 2024–2025 submissions and final decisions obtained via the `OpenReview API-V2` under CC BY 4.0; our crawl, de-identification, and parsing pipeline and the rules for domain labeling and class balancing are described in App. C and summarized in Table 6.

**Models & training.**    Exact model checkpoints and modalities appear in Sec. 4.2. The prompt template and label verbalizers are given in App. D; the continual pre-training corpus construction, packing block size, and optimization details are in Sec. 3.1 and App. E. All experiments were run on a two `NVIDIA A100 80GB` GPUs; precision and optimizer choices match App. E.4.

**Baselines & features.**    The 29 engineered features and model choices are documented in Sec. 4.5 and App. H. Evaluation. We report Accuracy, Macro-Precision, Macro-Recall and F1 with results in Tables 1–2, 8-10; the out-of-ditribution tests is detailed in Apps. G. The confidence-stratified utility analysis includes formulas and binning definitions in Sec. 5, Apps. I- J.

Upon publication, we will release our complete codebase and processed datasets, with rebuild scripts, to facilitate replication and extension of this work.

## B    USE OF LARGE LANGUAGE MODELS

In this work, we used large language models (LLMs) for two distinct purposes. First, we employed OpenAI's ChatGPT (GPT-5) exclusively for grammar correction and improving the fluency of the manuscript. Second, we evaluated ChatGPT's performance on our prediction task as part of the experimental analysis. Significantly, the model did not contribute to the research design, methodology, or interpretation of results; its role in writing was strictly limited to polishing sentence structure and enhancing readability. All technical contributions remain the sole work of the authors.

## C    DATA COLLECTION AND PRE-PROCESSING

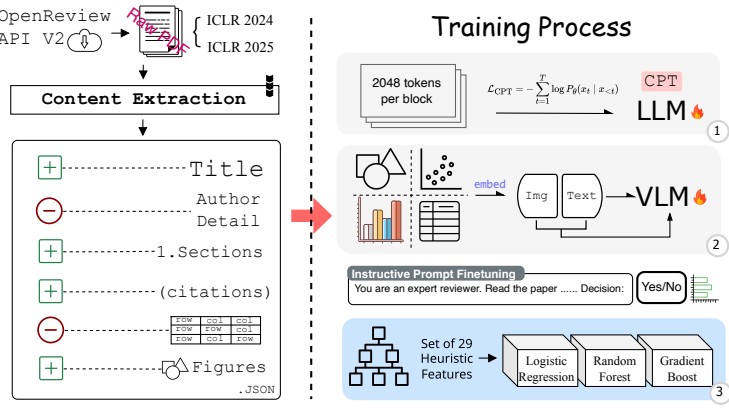

Figure 5: Data collection and preprocessing workflow and training pipeline.

We collect all ICLR 2025 and 2024 submissions and their corresponding final decisions (*accepted* or *rejected*) via the `OpenReview API-V2`. All acquired data complies with the Creative Commons Attribution 4.0 International (CC BY 4.0) license. The papers were further partitioned into four main subfields based on title keywords: Large Language Models (LLM), Computer Vision (CV), Reinforcement Learning (RL), and Theoretical (Theory). Papers that do not fall into these categories are left for future discussion. Summary counts for each subfield are reported in Table 6.

From the collected submissions, we construct two distinct datasets for our analysis: a complete ICLR2025 dataset as well as balanced domain-specific datasets by combining papers from both

ICLR 2024 and 2025 to ensure equal representation of accepted and rejected papers, addressing potential class imbalance issues that could bias our analysis.

We employ MINERU (Wang et al., 2024), an OCR-based tool, to extract structured content from the collected PDFs. As shown in Figure 5.

MINERU processes each document by separating text, images, tables, and equations, and generates a structured `JSON` representation. From this output, we retain only elements labeled as figures, tables, or equations, and restricted text extraction to the title, abstract, and introduction sections for use in our prediction model.[4] The final representation for each paper consisted of clean text files for the targeted sections, alongside organized visual elements paired with their original captions.

| Domain | Imbalanced | | | Balanced | | |
|---|---|---|---|---|---|---|
| | Size | Accept | Reject | Size | Accept | Reject |
| LLM | 3,716 | 1,253 | 2,463 | 3,238 | 1,619 | 1,619 |
| CV | 2,779 | 951 | 1,828 | 3,520 | 1,760 | 1,760 |
| RL | 1,253 | 440 | 813 | 1,526 | 763 | 763 |
| Theory | 1,741 | 625 | 1,116 | 1,974 | 987 | 987 |
| Combined | 9,489 | 3,269 | 6,220 | 10,258 | 5,129 | 5,129 |
| All | 11,601 | 4,000 | 7,601 | 10,258 | 5,129 | 5,129 |

Table 6: Data distribution across four domains for both the imbalanced and balanced datasets

## D    PROMPT TEMPLATE

We design an instructive prompt template that presents the paper's features within a natural-language query and guides the model to generate a decision token corresponding to one of the two target classes: *accept* or *reject*.

---

**Input Example**

**Template** $\mathcal{T}(x)$:
```
You are an expert reviewer. Read the paper content and decide if it
should be accepted.
Paper content: ⟨x⟩
Decision:
```

---

Given a paper input instance $x$ and prompt template $\mathcal{T}(x)$, the model defines a conditional probability over the *label verbalizer* Tam et al. (2021).

## E    CONTINUAL PRE-TRAINING

### E.1    MOTIVATION

Continual pre-training (CPT) adapts a strong general-purpose language model to the peer-review domain by further training on large-scale, unlabeled scientific corpora. Unlike supervised fine-tuning, CPT retains the original causal language modeling objective, thereby aligning the model's generative priors with the linguistic and structural regularities of academic manuscripts. This is particularly important in OpenReviewer, where downstream tasks rely on prompt-conditioned generation rather than explicit classification heads. This section will describe the training detail used for CPT.

### E.2    INPUT SETTING

We construct the CPT corpus by aggregating unlabeled texts from academic paper PDFs processed with MinerU. Each document is concatenated with an EOS separator, tokenized using the model's native tokenizer, and packed into fixed-length blocks of size $B$ (default $B = 2048$). This block-packing strategy eliminates under-filled sequences and ensures efficient utilization of training batches. The input IDs and labels are identical, enabling pure causal next-token prediction.

---

[4]Manual spot-checking confirmed high quality of the extracted content.

This preprocessing not only exposes the model to scientific writing styles, rhetorical markers, and citation format, etc. but also reduces the domain gap between generic pre-training corpora and the specialized peer-review domain.

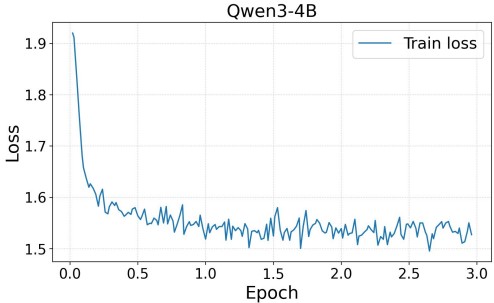 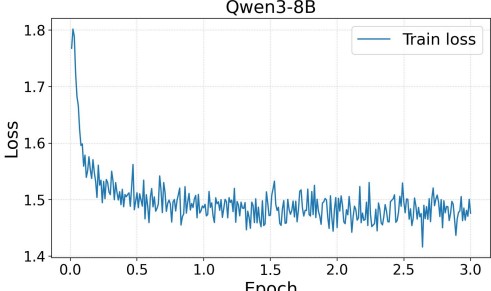

Figure 6: Continual pre-training loss results of Qwen3-4B and Qwen3-8B on the balanced dataset corpus.

### E.3 RESULT

During continual pre-training on the balanced corpus, the training loss decreases steadily for both Qwen3-4B and Qwen3-8B, indicating stable optimization. The 8B model converges slightly faster and to a lower final loss than the 4B model, consistent with its larger capacity. We observed no signs of divergence or instability across the three epochs, suggesting CPT effectively adapts the base models to scientific writing before downstream fine-tuning.

### E.4 HYPERPARAMETER SETTINGS

To maintain training stability, we adopt AdamW optimization with cosine learning rate decay, gradient checkpointing, and norm clipping. CPT is performed *prior* to prompt-based fine-tuning so that the updated parameters $\theta$ encode domain knowledge without introducing task-specific biases. Training was conducted on a single NVIDIA A100 80GB GPU.

| Hyperparameter | Value |
| --- | --- |
| Backbone Model | Qwen3-4B(8B) |
| Sequence Length ($B$) | 2048 |
| Batch Size (per device) | 2 |
| Gradient Accumulation | 8 (effective batch = 2×8×GPUs) |
| Epochs | 3 |
| Learning Rate | $1(2) \times 10^{-5}$ |
| Warmup Ratio | 0.1 |
| Weight Decay | 0.1 |
| Optimizer | AdamW |
| Scheduler | Cosine decay |
| Precision | bfloat16 (default) |
| Attention Backend | SDPA (FlashAttention-2 optional) |
| Gradient Checkpointing | Enabled |
| Max Grad Norm | 1.0 |

Table 7: Hyperparameter settings for continual pre-training in OpenReviewer.

## F  ADDITIONAL RESULTS ON VL-MODEL

### F.1  RESULTS ON IMBALANCE DATASET (IN-DISTRIBUTION)

Table 8 shows results on the imbalanced dataset. The models exhibit base-rate and threshold bias: minimizing loss encourages predicting the majority class. The prediction becomes more sensitive to textitreject patterns while under-covering the minority.

*Imbalanced In-Distribution Test*

| SUB-DOMAIN | LLM | | | | CV | | | | RL | | | | THEORY | | | | ALL | | | |
|---|---|---|---|---|---|---|---|---|---|---|---|---|---|---|---|---|---|---|---|---|
| | ACC | MAC-P | MAC-R | F1 | ACC | MAC-P | MAC-R | F1 | ACC | MAC-P | MAC-R | F1 | ACC | MAC-P | MAC-R | F1 | ACC | MAC-P | MAC-R | F1 |
| txt&img* (Qwen-VL-3B) | 35.7 | 17.9 | 50.0 | 26.3 | 35.6 | 17.8 | 50.0 | 26.3 | 32.9 | 16.5 | 50.0 | 24.8 | 33.1 | 16.6 | 50.0 | 24.9 | 34.8 | 17.4 | 50.0 | 25.8 |
| txt* | 35.7 | 17.9 | 50.0 | 26.3 | 35.7 | 17.8 | 50.0 | 26.3 | 32.9 | 16.5 | 50.0 | 24.8 | 33.1 | 16.6 | 50.0 | 24.9 | 34.8 | 17.4 | 50.0 | 25.8 |
| txt&img | 73.2 | 70.9 | 71.2 | 71.0 | 63.1 | 63.5 | 64.0 | 59.9 | 59.6 | 55.4 | 55.7 | 55.5 | 68.5 | 63.1 | 59.3 | 59.4 | 74.8 | 65.8 | 76.9 | 66.4 |
| txt | 74.2 | 68.2 | 69.5 | 68.2 | 69.5 | 68.1 | 69.5 | 68.2 | 66.3 | 57.9 | 53.9 | 51.9 | 67.2 | 62.5 | 62.0 | 62.2 | 75.5 | 71.9 | 69.4 | 67.1 |
| txt&img* (Gemma-3-4B) | 35.6 | 17.8 | 49.8 | 26.2 | 35.6 | 17.8 | 50.0 | 26.3 | 32.5 | 16.3 | 49.4 | 24.5 | 33.1 | 16.6 | 50.0 | 24.9 | 34.4 | 17.2 | 50.0 | 25.6 |
| txt* | 34.0 | 17.0 | 50.0 | 25.4 | 34.6 | 17.3 | 50.0 | 25.7 | 34.8 | 17.4 | 50.0 | 25.8 | 37.1 | 18.6 | 50.0 | 27.1 | 34.4 | 17.2 | 50.0 | 25.6 |
| txt&img | 64.3 | 32.2 | 50.0 | 39.1 | 63.8 | 54.4 | 50.9 | 43.7 | 67.9 | 83.8 | 51.3 | 42.8 | 66.0 | 55.9 | 55.0 | 45.6 | 76.8 | 91.5 | 35.7 | 51.4 |
| txt | 70.7 | 71.4 | 59.1 | 57.8 | 73.7 | 72.2 | 66.5 | 67.4 | 69.6 | 66.3 | 65.7 | 65.9 | 72.5 | 72.5 | 66.1 | 66.7 | 72.4 | 70.2 | 64.4 | 65.1 |

* Baseline models.

Table 8: Accuracy performance (%) of `Qwen2.5-VL-3B-Instruct` and `Gemma-3-4B-it` across four broad domains on imbalanced dataset. Mac-P and Mac-R denote Macro Precision and Macro Recall, respectively.

### F.2  MORE QUALITY ANALYSIS

Figure 7 shows additional examples of these two patterns from prior submissions (Kang & Oh, 2025; Zhang et al., 2024; Zhou et al., 2024; Feng et al., 2024; Ruan et al., 2025; Huynh et al., 2025). Most image-help cases are teaser images, which usually contain clear text and visual cues that support the model's judgment. In contrast, many image-mislead cases come from result analysis figures rather than teaser images, and thus contain little or no explicit textual guidance, making them harder for the model to interpret correctly.

**Case 1 Image Helps:** Text **Incorrect**, Text + Image **Correct**.

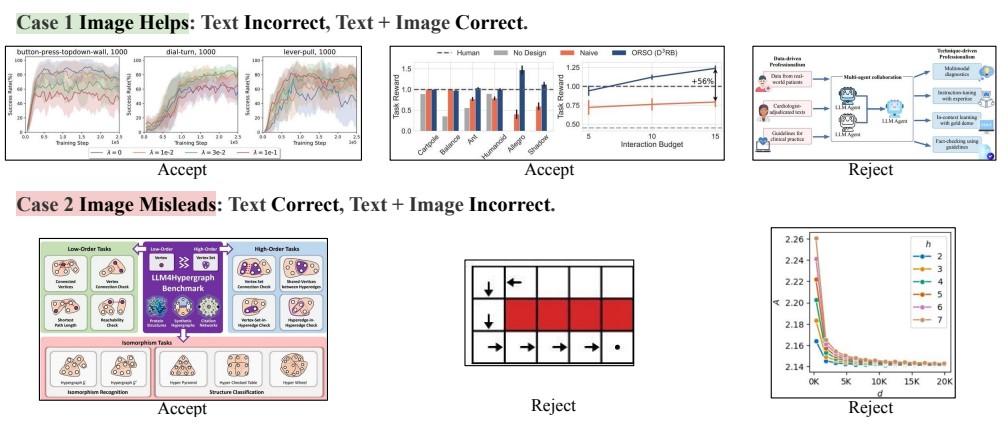

**Case 2 Image Misleads:** Text **Correct**, Text + Image **Incorrect**.

Figure 7: More examples of vision–language model predictions on previous submission.

### F.3  IMAGE-ONLY FOR PREDICTION

We further evaluate models using images as the sole input modality. First, we employ `DINO-v2` (Oquab et al., 2023)[5] as a classifier, where the inputs are the first two main figures from each paper. This setting yields an accuracy of 39.5% and an F1 score of 49.8%. In addition, we experiment with converting the first two pages of each PDF into images and training `Qwen-VL` with these image-only inputs. However, the performance in this setting remains close to that of the untrained baseline.

---

[5]https://huggingface.co/facebook/dinov2-base

## G  ABLATION STUDIES

We train on the imbalanced ICLR-2025 split and evaluate on a balanced 50/50 test to probe robustness. Across sizes, OOD accuracy hovers around 67–69%, with noticeable drops in macro-recall/F1 versus in-distribution, reflecting a reject-majority bias learned from imbalanced training. CPT offers modest, inconsistent gains (slightly higher macro-recall/F1 in some domains) but does not eliminate the bias; larger models (8B) do not guarantee better OOD generalization than 4B. Overall, results show that class balance during training matters more than scale, and that simple fine-tuning on imbalanced data leads to systematic under-coverage of ACCEPT, suggesting the need for rebalancing, threshold calibration, or post-hoc confidence conditioning for reliable deployment.

*LLMs Out-of-Distribution Test*

| SUB-DOMAIN | ALL | | | | LLM | | | | CV | | | | RL | | | | THEORY | | | |
|---|---|---|---|---|---|---|---|---|---|---|---|---|---|---|---|---|---|---|---|---|
| | ACC | MAC-P | MAC-R | F1 | ACC | MAC-P | MAC-R | F1 | ACC | MAC-P | MAC-R | F1 | ACC | MAC-P | MAC-R | F1 | ACC | MAC-P | MAC-R | F1 |
| CPT 4B | 67.8 | 72.5 | 67.0 | 68.4 | 65.6 | 71.4 | 64.9 | 65.6 | 68.2 | 70.8 | 69.9 | 71.6 | 66.4 | 68.1 | 53.1 | 46.7 | 54.8 | 54.3 | 51.9 | 44.2 |
| CPT 8B | 68.5 | 68.2 | 70.5 | 70.2 | 67.9 | 68.3 | 70.3 | 69.1 | 68.7 | 72.9 | 70.1 | 71.8 | 65.6 | 62.2 | 52.4 | 54.3 | 53.7 | 48.8 | 51.2 | 44.2 |
| Orig 4B | 68.6 | 70.9 | 69.0 | 70.8 | 65.9 | 67.4 | 66.8 | 68.0 | 67.5 | 71.5 | 68.4 | 69.8 | 62.8 | 67.4 | 62.7 | 62.3 | 59.5 | 65.5 | 58.5 | 55.8 |
| Orig 8B | 67.0 | 70.5 | 66.1 | 67.4 | 66.6 | 67.2 | 68.3 | 66.7 | 67.1 | 70.8 | 67.9 | 69.3 | 61.6 | 64.8 | 59.7 | 57.2 | 52.3 | 51.3 | 50.4 | 44.6 |

Table 9: Ablation results on the imbalanced ICLR 2025 dataset. Models are trained with the original accept/reject ratio (31.7% / 68.3%) and evaluated on the balanced 50/50 Out-of-Distribution test set.

*VLMs Out-of-Distribution Test*

| SUB-DOMAIN | ALL | | | | CV | | | | RL | | | | THEORY | | | | LLM | | | |
|---|---|---|---|---|---|---|---|---|---|---|---|---|---|---|---|---|---|---|---|---|
| | ACC | MAC-P | MAC-R | F1 | ACC | MAC-P | MAC-R | F1 | ACC | MAC-P | MAC-R | F1 | ACC | MAC-P | MAC-R | F1 | ACC | MAC-P | MAC-R | F1 |
| Qwen txt&img | 75.2 | 89.8 | 55.4 | 68.5 | 66.7 | 63.7 | 75.6 | 69.2 | 59.9 | 67.8 | 45.5 | 54.4 | 59.1 | 73.8 | 44.5 | 47.0 | 65.0 | 83.2 | 45.1 | 58.5 |
| Qwen txt | 76.2 | 89.4 | 58.0 | 70.3 | 75.2 | 80.7 | 65.7 | 72.4 | 55.7 | 88.9 | 18.2 | 30.2 | 66.1 | 82.0 | 45.6 | 58.6 | 59.3 | 87.5 | 29.9 | 44.6 |
| Gemma txt&img | 70.2 | 80.2 | 69.5 | 67.0 | 53.2 | 73.7 | 8.1 | 14.7 | 50.3 | 58.1 | 20.5 | 30.3 | 60.2 | 58.7 | 82.2 | 68.5 | 62.0 | 64.4 | 68.3 | 66.3 |
| Gemma txt | 76.2 | 81.2 | 75.7 | 75.0 | 53.7 | 58.0 | 23.3 | 33.2 | 57.5 | 63.5 | 45.5 | 53.0 | 54.4 | 73.1 | 21.1 | 32.8 | 55.7 | 75.4 | 28.1 | 40.9 |

Table 10: Accuracy performance (%) of `Qwen2.5-VL-3B-Instruct` and `Gemma-3-4B-it` across four broad domains under the Out-of-Distribution Test setting

## H  COMPLETE TABLE OF WHITE BOX FEATURES

The complete 29 white-box features importance are reported in Table 11.

## I  $R_2$ DERIVATION

We quantify how reliability scales with certainty by *separately for each predicted class* (ACCEPT, REJECT) fitting an ordinary least squares line to bin-level precision vs. confidence (using the *filtered bin midpoints* as $x$):

Given paired points $\{(x_i, y_i)\}$ where $x_i$ is the confidence-bin midpoint and $y_i$ the corresponding precision:

$$\text{Fit:} \quad y = mx + b, \tag{8}$$

$$\text{Residual sum of squares:} \quad SS_{\text{res}} = \sum_i (y_i - \hat{y}_i)^2, \tag{9}$$

$$\text{Total sum of squares:} \quad SS_{\text{tot}} = \sum_i (y_i - \bar{y})^2, \tag{10}$$

$$\text{Coefficient of determination:} \quad R^2 = 1 - \frac{SS_{\text{res}}}{SS_{\text{tot}}}. \tag{11}$$

**Interpretation.**

- $R^2 = 1$: perfect linear fit.

| Feature | Bal | Imb |
|---|---|---|
| **Structure** | | |
| total words | 0.0792 | 0.0739 |
| total pages | 0.0658 | 0.0575 |
| header count | 0.0569 | 0.0587 |
| section balance variance | 0.0474 | 0.0492 |
| words/page | 0.0446 | 0.0437 |
| **Visual Content** | | |
| avg caption length | 0.0439 | 0.0465 |
| image density | 0.0374 | 0.0378 |
| table density | 0.0363 | 0.0370 |
| image count | 0.0362 | 0.0332 |
| equation density | 0.0360 | 0.0372 |
| table count | 0.0355 | 0.0356 |
| equation count | 0.0315 | 0.0341 |
| **Citation Engagement** | | |
| citations in text | 0.0448 | 0.0403 |
| citation density | 0.0408 | 0.0395 |

(a) Structural, visual, and citation features

| Feature | Bal | Imb |
|---|---|---|
| **Methodological Rigor** | | |
| dataset mentions | 0.0389 | 0.0374 |
| metrics mentions | 0.0355 | 0.0356 |
| baseline mentions | 0.0241 | 0.0240 |
| statistical tests | 0.0061 | 0.0058 |
| experiment count | 0.0030 | 0.0031 |
| **Writing Quality** | | |
| abstract word count | 0.0430 | 0.0446 |
| avg sentence length | 0.0423 | 0.0432 |
| **Novelty & Contribution** | | |
| novel method claims | 0.0346 | 0.0365 |
| comparison studies | 0.0327 | 0.0356 |
| contribution statements | 0.0318 | 0.0314 |
| **App. Material** | | |
| word count (appendix) | 0.0188 | 0.0203 |
| header count (appendix) | 0.0179 | 0.0177 |
| images count (appendix) | 0.0125 | 0.0134 |
| table count (appendix) | 0.0121 | 0.0161 |
| equation count (appendix) | 0.0088 | 0.0095 |

(b) Methodological, writing, novelty, and appendix features

*Bal* = balanced dataset importance; *Imb* = imbalanced dataset importance.

Table 11: Feature importance across balanced (Bal) and imbalanced (Imb) datasets using a Random Forest. Values are normalized importances.

- $R^2 = 0$: no better than predicting the mean $\bar{y}$.
- $R^2 < 0$: worse than predicting the mean.
- $R^2 = $ NaN: not enough points, constant $x$, or zero variance in $y$ ($SS_{\text{tot}} = 0$).

## J  LINEAR PRECISION-CONFIDENCE RELATIONSHIP

We further quantify how reliability scales with certainty by fitting, separately for ACCEPT and REJECT, an least squares model of precision against confidence, and we report the linear coefficient of determination $R^2$ in the figures to characterize the strength of the linearity.

On the **balanced** dataset (Fig. 4a, 4b), two classes exhibit similar $R^2$ values, indicating that increases in confidence translate into nearly equivalent gains in precision for both ACCEPT and REJECT. Moreover, the Qwen3-4B model exhibits a stronger linear relationship than the Qwen3-8B model on this dataset, with the highest fit $R^2 = 0.85$.

On the **imbalanced** dataset (Fig. 4c, 4d), by contrast, the precision–confidence relationship diverges across classes: the minority class (ACCEPT) typically shows a lower $R^2$, reflecting weaker separability than under balanced training.

