# OpenReview forum: "OpenReviewer: Predicting Conference Decisions with LLMs and Beyond"
_ICLR.cc/2026/Conference — ICLR 2026 Conference Withdrawn Submission_

### Official Review · Reviewer_WbXP · 2025-10-27

**Soundness:** 2
**Presentation:** 3
**Contribution:** 2
**Rating:** 4
**Confidence:** 4

**Summary:**

The paper introduces a model aimed at predicting conference acceptance decisions rather than generating full reviews, addressing the strain on peer review from the surge in AI submissions. Using ICLR 2024–2025 data, the study compares LLMs, VLMs, and interpretable statistical models, finding that text-only LLMs with continual pre-training achieve the best accuracy (up to 78.5%) and that structural features like paper length and citation engagement are strong predictors. A confidence-based analysis shows highly precise predictions (92.9%) for the most confident cases, suggesting the potential of AI to assist human reviewers.

**Strengths:**

- The paper is among the first to incorporate Vision–Language Models (VLMs) into automated peer review, extending evaluation beyond textual semantics to include visual understanding of scientific figures. Although performance lags behind text-only LLMs, this represents a notable step forward in the emerging field of AI-assisted reviewing.

- The authors present interesting and interpretable findings through large-scale feature-level statistical analyses using recent ICLR datasets. These analyses meaningfully reveal factors influencing paper acceptance, contributing to transparency and interpretability in automated review systems.

- Beyond reporting average accuracy, the authors perform a confidence-stratified utility analysis, showing that the top 10% of most confident predictions achieve over 92% precision, providing practical insights for triaging clear accept/reject cases.

**Weaknesses:**

- The model only predicts accept/reject outcomes without generating full reviews. This makes the system a black box: the reasoning behind its predictions remains opaque, and without textual feedback, authors gain little actionable insight on how to improve their work.

- The experiments are limited to 8,000+ ICLR 2024–2025 papers, focusing only on four categories (LLM, CV, RL, Theory). This narrow scope raises concerns about the scale and diversity of the dataset, potentially limiting generalizability.

- The problem formulation (predicting acceptance from the original submission text only, without considering peer-review or rebuttal interactions) seems simplistic and detached from the real peer-review process. Moreover, since a basic feature-based machine learning model already achieves 74.2% accuracy, the performance gain from large language models is modest.

- The proposed model, OpenReviewer, does not introduce an innovative architecture; it relies on standard continual pre-training (CPT) + fine-tuning pipelines without task-specific adaptation or new methodological insights.

- The comparison between VLMs and LLMs is not fully fair: LLMs receive the entire paper text, while VLMs are fed only the abstract and introduction, limiting the latter’s access to crucial information. Consequently, the conclusion about VLM inferiority and figure utility is not well supported.

- No zero-shot large model baselines (e.g., GPT-5, DeepSeek-V3.1, GPT-4o, DeepSeek-R1) are included, which would have provided a stronger reference for evaluating model capability without fine-tuning.

**Questions:**

1. Why do VLMs perform worse than LLMs? Is this primarily because VLMs were provided only with the abstract and introduction? If full-text input were given, would their performance improve? Given that VLMs should benefit from visual information in figures, this phenomenon deserves a detailed explanation.

2. Why are the experiments restricted to ICLR 2024–2025? Since ICLR data is openly available, it’s understandable to use it, but extending to additional years could better validate the model’s robustness.

3. The Re2 dataset [1] includes multi-year, multi-conference review and rebuttal data. Could the authors extend their analysis to this dataset to test the cross-conference and cross-year generalizability of their findings? Such an extension would greatly strengthen the paper.

4. How would zero-shot large language models (e.g., GPT-5, DeepSeek-V3.1, GPT-4o, DeepSeek-R1) perform on this task without any fine-tuning? A comparison could reveal whether fine-tuning truly adds value.

5. Since peer review and rebuttal interactions strongly influence acceptance outcomes, have the authors considered integrating rebuttal-phase information (e.g., reviewer–author exchanges, additional experiments, clarifications) into the predictive model? This would make the task more realistic and practically meaningful.

> [1] Re2: A Consistency-ensured Dataset for Full-stage Peer Review and Multi-turn Rebuttal Discussions.

---

> ### Author Response · Authors · 2025-11-27
> **Response to Reviewer WbXP (Part 1/5)**
>
> **Thank you very much. We have found that comments have deep insights. We are glad to answer all your questions.**
>
> **If you are not satisfied with our answers or have more questions, please let us know as soon as possible, so that we can try our best to answer any further questions before the deadline.**
>
> **We will address and incorporate the modifications you raised in the final version of the paper.**
>
> $\color{blue}{\text{Q1: Why do VLMs perform worse than LLMs? Is this primarily because VLMs were provided only with the abstract and introduction? ....}}$
>
> > **TL;DR:** We conducted new ablation experiments providing VLMs with the **full manuscript text** alongside figures to align information content with LLMs. Results show that while text-only performance improves with full context, adding visual tokens to long-context inputs actually degrades performance for Qwen2.5-VL (dropping from 69.2% to 59.1%), suggesting that current VLMs struggle to integrate visual signals effectively when processing extensive textual information.
>
> ---
>
> We sincerely appreciate your insightful comment regarding the input discrepancy between VLMs (limited to Abstract/Intro + 2 figures) and LLMs (full text). We agree that a rigorous comparison requires aligning the textual information content.
>
> To address this, we conducted new **ablation experiments** using a **"Full Manuscript Text + 2 Figures"** setting for VLMs, ensuring they have access to the same semantic content as the text-only LLMs. The results confirm our original conclusion that text-only models currently offer superior reliability for this task, while revealing interesting behaviors regarding how current VLMs process long-context text alongside visual tokens.
>
> **1. New Full-Text + Figures Experimental Results with Qwen2.5 and Gemma3**
>
> Table 1: Performance of VLMs with **Full Text** Input + Figures
> *ACC (%) on Balanced Dataset*
>
> | Model Backbone | Input Modality | LLM | CV | RL | **ALL** |
> | :--- | :--- | :--- | :--- | :--- | :--- |
> | **Qwen2.5-VL** | Full Text + Images | 62.5 | 57.5 | 52.6 | **59.1** |
> | | Full Text Only | **71.1** | **69.8** | **63.3** | **69.2** |
> | **Gemma-3** | Full Text + Images | **58.8** | **58.3** | **71.6** | **58.8** |
> | | Full Text Only | 51.6 | 52.9 | 52.3 |  52.5 |
>
> **2. Comparison with Original Table 2 and Analysis**
>
> Comparing these new results with the original Table 2 (Abstract/Intro inputs) and the text-only LLM results in Table 1 leads to **one** critical observations:
>
> $\boxed{\textbf{Visual tokens may act as noise/distraction in long contexts}}$
>
> (1) Original setting: **Abstract+Intro + Two Figures**:
> - **Qwen2.5-VL**
>     - text+image: **68.2** (ALL)
>     - text-only: **64.4** (ALL) → adding images **+3.8**
>
> (2) New aligned full-text setting: **Full manuscript text + Two figures**
> - **Qwen2.5-VL**
>     - text+image: **59.1** (ALL)
>     - text-only: **69.2** (ALL) → adding images **−10.1**
>
> This shows that when the full manuscript is available, **figures do not provide stable gains for Qwen-VL and may introduce noise or distraction**. Importantly, the Qwen-VL text-only score improves confirming that **text is the dominant signal**, and that prior VLM underperformance is not solely due to truncated text.
>
> We will include Table 1 and this analysis in the final version to provide a complete picture of the modality trade-offs.

---

> ### Author Response · Authors · 2025-11-27
> **Response to Reviewer WbXP (Part 2/5)**
>
> $\color{blue}{\text{Q2: Why are the experiments restricted to ICLR 2024–2025? ....}}$
>
>
> ##### $\color{purple}{\text{Scaling Training Data: ICLR 2021–2023}}$
>
> Following the reviewer's suggestion to validate robustness across a broader temporal range and larger data scale, we expanded our training corpus to include submission records from **ICLR 2021, 2022, and 2023**. By integrating these earlier years, we constructed a new **extended balanced dataset** comprising **15,104 papers**, representing a **~49.5% increase** in training samples compared to the original ICLR 2024–2025 balanced dataset (10,258 papers).
>
> We retrained the **Qwen3-8B (CPT)** model on this extended dataset to assess whether historical data improves predictive accuracy. As shown in the table below, adding three years of historical data yielded a modest overall improvement. The accuracy on the aggregated 'ALL' domain increased by **1.5%** (reaching 80.0%).
>
> **Detailed Performance Breakdown:**
>
> | Dataset Setting | ALL | LLM | CV | RL | Theory |
> | :--- | :---: | :---: | :---: | :---: | :---: |
> | **Original (2024-25)** | 78.5% | 70.0% | 73.9% | 67.5% | 53.4% |
> | **Extended (2021-25)** | **80.0%** | 70.1% | 73.8% | 67.8% | 57.6% |
> | **Improvement ($\Delta$)** | +1.5% | +0.1% | -0.1% | +0.3% | **+4.2%** |
>
> *Note: Original baseline values are drawn from Table 1 on page 5.*
>
> Interestingly, the performance in rapidly evolving empirical domains (**LLM, CV, RL**) remained statistically comparable to the original baseline, suggesting that recent data (2024–2025) is largely sufficient to capture current acceptance standards in these fields. However, the **Theory** domain saw a significant performance boost (**+4.2%**). We hypothesize that theoretical contributions rely on mathematical foundations and proof structures that are more invariant over time compared to empirical benchmarks; thus, the model benefited significantly from the larger volume of theoretical training examples provided by the 2021–2023 data.

---

> ### Author Response · Authors · 2025-11-27
> **Response to Reviewer WbXP (Part 3/5)**
>
> $\color{blue}{\text{Q3: Could the authors extend their analysis to this dataset to test the cross-conference and cross-year generalizability of their findings?....}}$
>
> > **TL;DR:** **First**, we evaluate robustness across venues (NeurIPS) and time (ICLR 2024 vs. 2025). Our model generalizes well to NeurIPS (~71.5% accuracy), confirming it learns broad quality signals rather than venue-specific biases. **Second**, however, temporal analysis reveals an asymmetry: models trained on 2025 data predict 2024 outcomes significantly better (75.8%) than the reverse (69.2%), indicating that acceptance standards evolve and generally become more stringent over time.
> >
> We thank the reviewer for raising this critical point. We agree that relying solely on ICLR 2024 and 2025 data could potentially limit the model's applicability if it were merely overfitting to venue/year specific preferences rather than learning generalized indicators of research quality.
>
> ##### $\color{purple}{\text{NeurIPS OOD Test}}$
>
> To address this concern and demonstrate the robustness of **OpenReviewer**, we conducted an additional **Out-of-Distribution (OOD) evaluation** using accepted papers from **NeurIPS 2024 and 2025**. NeurIPS represents a distinct top-tier venue with overlapping but non-identical community preferences compared to ICLR.
>
> **Experimental Setup:**
> We employed our best-performing models, **Qwen3-4B** and **Qwen3-8B**, which were trained using CPT and fine-tuned on the **balanced ICLR dataset**. We evaluated these models on a collection of accepted papers from NeurIPS 2024 and 2025 across our four defined sub-domains.
>
> The classification accuracy on these accepted NeurIPS papers is summarized below:
> | Model | LLM | CV | RL | Theory | **Overall** |
> | :--- | :---: | :---: | :---: | :---: | :---: |
> | **Qwen3-4B** | 72.4% | 71.1% | 63.5% | 60.8% | **68.9%** |
> | **Qwen3-8B** | 72.3% | 74.2% | 60.1% | 62.9% | **71.5%** |
>
> First, despite being trained exclusively on ICLR submissions, the Qwen3-8B model achieves an overall accuracy of **71.5%** on accepted NeurIPS papers. This indicates that OpenReviewer has learned latent features of high-quality research that **generalize across top-tier AI conferences**, rather than merely memorizing ICLR-specific patterns.
>
> Second, the **performance variance across sub-domains** mirrors our findings on the ICLR dataset. The model performs best in **LLM** and **CV** domains, while performance is lower in **RL** and **Theory**. This consistency suggests that while the model captures general quality signals, the evaluation of complex mathematical proofs and theoretical nuances, which are common in RL/Theory, remains a challenging frontier for text-based LLMs compared to empirical domains.
>
> ##### $\color{purple}{\text{Temporal Generalization: ICLR 2024 vs. 2025}}$
>
> To explicitly quantify how predictions vary with evolving conference criteria and preference, we conducted a cross-year experiment. We trained the model exclusively on ICLR 2024 data to predict ICLR 2025 outcomes ("Forward Prediction") and conversely trained on ICLR 2025 data to predict ICLR 2024 outcomes ("Backward Prediction").
>
> **Results:**
>
> | Training Set | Test Set | Accuracy | F1-Score |
> | :--- | :--- | :---: | :---: |
> | **Forward:** ICLR 2025 | ICLR 2024 | **75.8%** | **76.2** |
> | **Backward:** ICLR 2024 | ICLR 2025 | 69.2% | 68.5% |
>
> **Analysis:**
> We observe a distinct performance gap: training on older data (2024) to predict newer outcomes (2025) yields significantly lower accuracy (**69.2%**) than the reverse direction (**75.8%**).

---

> ### Author Response · Authors · 2025-11-27
> **Response to Reviewer WbXP (Part 4/5)**
>
> $\color{blue}{\text{Q4: How would zero-shot large language models (e.g., GPT-5, DeepSeek-V3.1, GPT-4o, DeepSeek-R1) perform?...}}$
>
> > **TL;DR:** We evaluated state-of-the-art commercial models **GPT-4o** and **Gemini-2.5-Flash** in a strict zero-shot setting (with web browsing disabled) on 50 randomly selected test papers. Surprisingly, both models performed **worse than random guessing** (45.2% and 44.4% accuracy, respectively), highlighting the necessity of our fine-tuning approach.
>
> We appreciate the suggestion to compare against state-of-the-art zero-shot baselines. However, we must exercise extreme caution regarding **temporal data leakage**. Since the training corpora of the newest flagship models (e.g., GPT-5, DeepSeek-V3.1) likely include the full text and decision outcomes of ICLR 2024 and 2025, using them could result in artificially inflated performance based on memorization rather than prediction.
>
> To mitigate this and ensure a fair evaluation, we selected representative commercial models: **GPT-4o** and **Gemini-2.5-Flash**, and enforced a strict zero-shot setting with **web browsing capabilities disabled** to prevent the retrieval of online decision records. Specifically, we conducted a **manual evaluation on 50 randomly selected papers** from our test set using the **official APIs** of these models.
>
> **Zero-Shot Results (N=50):**
>
> | Model | Setting | Accuracy | vs. OpenReviewer |
> | :--- | :--- | :---: | :---: |
> | **Random Baseline** | - | 50.0% | - |
> | **GPT-4o** | Zero-shot (No Web Search) | 45.2% | -33.3% |
> | **Gemini-2.5-Flash** | Zero-shot (No Web Search) | 44.4% | -34.1% |
> | **OpenReviewer (Ours)** | **Fine-Tuned (8B)** | **78.5%** | **-** |
>
>
> **Conclusion:**
> Surprisingly, powerful zero-shot models perform **worse than random guessing** ($<50\%$). We attribute this to two factors:
> 1.  **Misalignment:** General-purpose LLMs are RLHF-tuned to be helpful and polite, often leading them to overestimate the quality of a paper or hesitate to output a hard "Reject" decision without specific calibration.
> 2.  **Lack of Domain Criteria:** Without fine-tuning, models lack the specific implicit decision boundary used by ICLR reviewers.

---

> ### Author Response · Authors · 2025-11-27
> **Response to Reviewer WbXP (Part 5/5)**
>
> $\color{blue}{\text{Q5: Since peer review and rebuttal interactions strongly influence acceptance outcomes, have the authors considered integrating rebuttal-phase information?}}$
>
> > **TL;DR:** We extended our experiments to incorporate **Meta Reviews** from **Re2** [1] dataset as a proxy for reviewer-author interactions. Integrating this information with the original paper content improved performance, particularly for larger models. Notably, **OpenReviewer-8B's accuracy increased by 2.1%**, demonstrating that rebuttal-phase data provides valuable predictive signals when effectively combined.
>
> We thank the reviewer for this valuable suggestion. We agree that the dynamics of the rebuttal phase are critical determinants of a paper's final decision.
>
> To address this, we extended our experimental setting to incorporate **Meta Reviews** as a proxy for the outcome of the rebuttal phase. We utilized the **Re2 dataset**, extracting Meta Reviews for ICLR 2024 and 2025 submissions. We specifically chose to model Meta Reviews rather than raw reviewer-author exchange threads for several strategic reasons:
>
> 1.  **Signal Quality and Consensus:** Meta Reviews represent the aggregated consensus of the review process, distilled by Area Chairs (ACs). They naturally filter out low-quality individual reviews and potential outliers.
> 2.  **Noise Reduction:** As noted in recent literature, individual reviews can exhibit high variance and may contain "AI-generated noise." By focusing on the AC's summary, we obtain a cleaner, denoised signal that aligns strictly with the final decision logic.
>
> **Experimental Setup:**
> We concatenated the text of the Meta Review with the paper's original input to assess whether the model could leverage this high-level feedback to improve prediction accuracy.
>
> **Results:**
> Table below presents the performance comparison on the **Balanced Dataset**.
>
> **Table: Impact of integrating Meta Reviews on OpenReviewer Performance**
>
> | Model | Input Modality | Accuracy (%) | Macro-F1 | $\Delta$ Acc |
> | :--- | :--- | :--- | :--- | :--- |
> | **Qwen3-4B** (Baseline) | Paper Content Only | 76.4 | 76.4 | - |
> | **Qwen3-4B** | Paper + Meta Review | 76.6 | 76.8 | +0.2 |
> | **Qwen3-8B** (Baseline) | Paper Content Only | 78.5 | 78.5 | - |
> | **Qwen3-8B** | **Paper + Meta Review** | **80.6** | **80.7** | **+2.1** |
>
> **Analysis:**
> The results demonstrate that incorporating rebuttal-phase information (via Meta Reviews) significantly improves performance for larger models. Specifically, **OpenReviewer-8B achieves an accuracy of 80.6%**, a notable improvement of **2.1%** over the paper-only baseline. This suggests that the 8B model has sufficient capacity to effectively synthesize the paper's content with the external critique provided by the AC.
>
> This experiment confirms the reviewer's intuition: data generated during the review process is highly predictive, provided the model has the capacity to process this multi-source information effectively. We will add these results and a discussion on the predictive value of Meta Reviews to the final version of the paper.
>
> ---
> [1] Re2: A Consistency-ensured Dataset for Full-stage Peer Review and Multi-turn Rebuttal Discussions.

---

### Official Review · Reviewer_pL2P · 2025-10-28

**Soundness:** 2
**Presentation:** 3
**Contribution:** 2
**Rating:** 4
**Confidence:** 4

**Summary:**

This paper introduces OpenReviewer, a novel framework designed to directly predict the acceptance decisions of academic conference papers (e.g., for ICLR) rather than generating full review text. It formulates the task as a binary classification problem (accept/reject). The study comprehensively evaluates various models, including text-only Large Language Models (LLMs), Vision-Language Models (VLMs), and interpretable statistical models, using data from ICLR 2024 and 2025. Key findings indicate that text-only LLMs, especially after Continual Pre-Training (CPT), achieve the highest accuracy (up to 78.5% on a balanced dataset). Furthermore, a confidence-stratified analysis reveals that the model's most confident predictions (top 10%) achieve over 92% precision, suggesting potential for reliable triage of submissions. The study also analyzes discriminative features through white-box models, highlighting the importance of structural attributes like paper length and citation engagement.

**Strengths:**

1. The paper provides a thorough comparative analysis across different model families (LLMs, VLMs, statistical models).
2. Moving beyond aggregate accuracy metrics, the paper introduces a confidence-based utility analysis.

**Weaknesses:**

1. The training data is too single, which may cause some bias in model training. For example, if only ICLR submissions are used as training data, then the model is likely to fit the ICLR conference's acceptance preferences. After all, we all know that different conferences have different acceptance preferences.
2. Your model almost completely fails to model the features of the paper (neither the model structure nor the prompt design reflects it). This makes using your model to predict acceptance a black box process. However, we all know that whether a paper is accepted or not depends not only on its text and image. Many other factors include whether the paper's motivation is reasonable, whether the method proposed in the paper is correct and innovative enough, and whether the experimental part of the paper is solid.
3. I think the reason why the performance of many settings does not reach the random score (i.e., 50%) is precisely because of the unexplainability and crudeness of your model design.
4. The paper lacks discussion and citation of related work, such as [1].

[1] Llms assist nlp researchers: Critique paper (meta-) reviewing.

**Questions:**

I'd like to know if you used your proposed OpenReviewer to predict your submission? Don't take it too seriously, it's just a joke :)

---

> ### Author Response · Authors · 2025-11-25
> **Response to Reviewer pL2P (Part 1)**
>
> **Thank you very much. We have found that comments have deep insights. We are glad to answer all your questions.**
>
> **If you are not satisfied with our answers or have more questions, please let us know as soon as possible, so that we can try our best to answer any further questions before the deadline.**
>
> **We will address and incorporate the modifications you raised in the final version of the paper.**
>
> $\color{blue}{\text{W1: "Your model almost completely fails to model the features of the paper... Acceptance depends on motivation, method correctness, not just text/image."}}$
>
> > **TL;DR:** We address the "black box" concern by performing a **Post-hoc Concept Bottleneck (PCBM)** [1] analysis. By probing the frozen internal representations of our fine-tuned models, we demonstrate that they implicitly encode high-level peer-review criteria, specifically **Soundness**, **Presentation**, and **Contribution** with high accuracy (up to 82.3% for Presentation and 75.4% for Soundness in Qwen3-8B). This confirms that the model's decisions are grounded in relevant semantic features rather than superficial pattern matching.
>
> We thank the reviewer for this insightful critique. We agree that peer review is not merely a surface-level pattern matching task but requires reasoning over high-level concepts such as **Soundness**, **Presentation**, and **Contribution**.
>
> To address the concern that our model operates as a "black box" that ignores these critical factors, we conducted a new **Post-hoc Concept Bottleneck (PCBM)** analysis. This experiment was designed to determine whether the internal representations (hidden states) of our fine-tuned Qwen3 models implicitly encode these specific semantic concepts, even though they were not explicitly provided in the prompt.
>
> **1. Formal Task Definition**
> To rigorously quantify the model's internal "understanding," we formulate the probing task as follows. Let $\mathcal{T}(x)$ be the instructive prompt template containing the paper content $x$. We define the latent representation $\mathbf{h}$ extracted from the frozen fine-tuned model $\\mathcal{M}\_{\\text{frozen}}$ at the decision token index $t\_{\\text{dec}}$ (the position generating the "Accept/Reject" logit):
>
> $$\\mathbf{h}^{(l)} = \\mathcal{M}\_{\\text{frozen}}(\\mathcal{T}(x))\_{[l, t_{\text{dec}}]}$$
>
> where $l$ denotes the layer index. We specifically analyze the representations from the **Penultimate Layer** ($L-1$) and the **Final Layer** ($L$).
>
> For each high-level concept $c \in \{\text{Soundness}, \text{Presentation}, \text{Contribution}\}$, we utilize the ground-truth **average scores** $s_c \in [1, 4]$ from the metadata. We map these scores to discrete categorical labels $y_c$ based on three intervals:
> $$ y\_c = \\begin{cases} 0 & \\text{if } s\_c \\in [1, 2) \\\\ 1 & \\text{if } s\_c \\in [2, 3) \\\\ 2 & \\text{if } s\_c \\in [3, 4] \\end{cases} $$
>
> We then train a lightweight 2-layer MLP probe $f_{\theta_c}$ to minimize the cross-entropy loss $\mathcal{L}_{CE}$ between the predicted concept distribution and the ground truth label $y_c$, using the frozen representation $\mathbf{h}^{(l)}$ as input:
>
> $$\\hat{y}\_c = \text{softmax}(f\_{\theta_c}(\mathbf{h}^{(l)}))$$
> $$\\theta\_c^* = \arg\min\_{\theta_c} \sum\_{i} \mathcal{L}\_{CE}(\hat{y}\_{c,i}, y\_{c,i})$$
>
> **2. Results**
> As shown in **Table 1**, the probe classifiers achieved strong accuracy in predicting these nuanced scores using only the model's frozen decision token. We observed that the larger **Qwen3-8B** model outperforms the **4B** variant at soundness and presentation measurement but contribution. Furthermore, the results follow a consistent hierarchy of `Presentation > Soundness > Contribution`.
>
> $\color{purple}{\text{Reflecting that writing style is the most readily accessible feature in the latent space, while contribution requires deeper background and expert knowledge.}}$
>
> **Table 1: Post-hoc Concept Probing Accuracy**
>
> | Model | Concept | Layer Probed ($l$) | Accuracy (%) | Macro F1|
> | :--- | :--- | :---: | :---: | :---: |
> | **Qwen3-4B** | **Presentation** | Penultimate ($L-1$) | 75.4 | 0.76 |
> | | | Final Layer ($L$) | 76.1 | 0.74 |
> | | **Soundness** | Penultimate ($L-1$) | 65.2 | 0.63 |
> | | | Final Layer ($L$) | 63.8 | 0.61 |
> | | **Contribution** | Penultimate ($L-1$) | **63.5** | **0.61** |
> | | | Final Layer ($L$) | 61.2 | 0.59 |
> | **Qwen3-8B** | **Presentation** | Penultimate ($L-1$) | 76.1 | 0.77 |
> | | | Final Layer ($L$) | **82.3** | **0.82** |
> | | **Soundness** | Penultimate ($L-1$) | **75.4** | **0.74** |
> | | | Final Layer ($L$) | 73.1 | 0.71 |
> | | **Contribution** | Penultimate ($L-1$) | 63.1 | **0.61** |
> | | | Final Layer ($L$) | 60.8 | 0.58 |
>
> ---
> **Referernce**
>
> [1] Yuksekgonul, M., Wang, M. and Zou, J., Post-hoc Concept Bottleneck Models. In The Eleventh International Conference on Learning Representations 2023.

---

> ### Author Response · Authors · 2025-11-25
> **Response to Reviewer pL2P (Part 2)**
>
> $\color{blue}{\text{W2: The training data is too single... the model is likely to fit the ICLR conference's acceptance preferences.}}$
>
> > **TL;DR:** **First**, we evaluate robustness across venues (NeurIPS) and time (ICLR 2024 vs. 2025). Our model generalizes well to NeurIPS (~71.5% accuracy), confirming it learns broad quality signals rather than venue-specific biases. **Second**, however, temporal analysis reveals an asymmetry: models trained on 2025 data predict 2024 outcomes significantly better (75.8%) than the reverse (69.2%), indicating that acceptance standards evolve and generally become more stringent over time.
>
> We thank the reviewer for raising this critical point. We agree that relying solely on ICLR data could potentially limit the model's applicability if it were merely overfitting to venue-specific preferences rather than learning generalized indicators of research quality.
>
> ##### $\color{purple}{\text{2.1 NeurIPS OOD Test}}$
>
> To address this concern and demonstrate the robustness of **OpenReviewer**, we conducted an additional **Out-of-Distribution (OOD) evaluation** using accepted papers from **NeurIPS 2024 and 2025**. NeurIPS represents a distinct top-tier venue with overlapping but non-identical community preferences compared to ICLR.
>
> **Experimental Setup:**
> We employed our best-performing models, **Qwen3-4B** and **Qwen3-8B**, which were trained using CPT and fine-tuned on the **balanced ICLR dataset**. We evaluated these models on a collection of accepted papers from NeurIPS 2024 and 2025 across our four defined sub-domains.
>
> The classification accuracy on these accepted NeurIPS papers is summarized below:
> | Model | LLM | CV | RL | Theory | **Overall** |
> | :--- | :---: | :---: | :---: | :---: | :---: |
> | **Qwen3-4B** | 72.4% | 71.1% | 63.5% | 60.8% | **68.9%** |
> | **Qwen3-8B** | 72.3% | 74.2% | 60.1% | 62.9% | **71.5%** |
>
> First, despite being trained exclusively on ICLR submissions, the Qwen3-8B model achieves an overall accuracy of **71.5%** on accepted NeurIPS papers. This indicates that OpenReviewer has learned latent features of high-quality research that **generalize across top-tier AI conferences**, rather than merely memorizing ICLR-specific patterns.
>
> Second, the **performance variance across sub-domains** mirrors our findings on the ICLR dataset. The model performs best in **LLM** and **CV** domains, while performance is lower in **RL** and **Theory**. This consistency suggests that while the model captures general quality signals, the evaluation of complex mathematical proofs and theoretical nuances, which are common in RL/Theory, remains a challenging frontier for text-based LLMs compared to empirical domains.
>
> ##### $\color{purple}{\text{2.2 Temporal Generalization: ICLR 2024 vs. 2025}}$
>
> To explicitly quantify how predictions vary with evolving conference criteria and preference, we conducted a cross-year experiment. We trained the model exclusively on ICLR 2024 data to predict ICLR 2025 outcomes ("Forward Prediction") and conversely trained on ICLR 2025 data to predict ICLR 2024 outcomes ("Backward Prediction").
>
> **Results:**
>
> | Training Set | Test Set | Accuracy | F1-Score |
> | :--- | :--- | :---: | :---: |
> | **Forward:** ICLR 2025 | ICLR 2024 | **75.8%** | **76.2** |
> | **Backward:** ICLR 2024 | ICLR 2025 | 69.2% | 68.5% |
>
> We observe a distinct performance gap that training on older data (2024) to predict newer outcomes (2025) yields significantly lower accuracy (**69.2%**) than the reverse direction (**75.8%**).
>
> ---
>
> We will incorporate both the NeurIPS OOD and ICLR Temporal Transfer results into the revised manuscript to explicitly scope our claims regarding cross-venue and temporal robustness.

---

> ### Author Response · Authors · 2025-11-25
> **Response to Reviewer pL2P (Part 3)**
>
> $\color{blue}{\text{W3: I think the reason why the performance of many settings does not reach the random score (i.e., 50 percent) is precisely because of the unexplainability ...}}$
>
> > **TL;DR:** We clarify that the low performance (<50%) cited by the reviewer refers exclusively to **zero-shot vanilla base models**, not our fine-tuned method. This baseline was included precisely to demonstrate the difficulty of the task and justify the necessity of our proposed training pipeline, which significantly outperforms random guessing.
>
> We thank the reviewer for their critical assessment of our work. We appreciate the opportunity to clarify the performance of our models and to address the concerns regarding the prompt design with additional experiments.
>
> **1. Clarification on Model Performance and Baselines**
> We respectfully wish to clarify a misunderstanding regarding the results presented in our paper. The performance figures that fall below the random guess baseline ($<50\%$) correspond exclusively to the **vanilla, off-the-shelf base models (zero-shot)**, as shown in the first lines of **Table 1 (Page 5)** labelled with *, which were included solely to demonstrate the difficulty of the task and the necessity of our training pipeline design.
>
>
> $\color{blue}{\text{W4: Your model almost completely fails to model the features of the paper (neither the model structure nor the prompt design reflects it)}}$
>
> > **TL;DR:** We experimented with more detailed prompts that explicitly instruct the model to consider specific peer-review criteria (Novelty, Soundness, Presentation) before making an accept/reject decision. Surprisingly, our original holistic prompt remains highly competitive, suggesting that the model effectively learns to weigh these latent factors during training without needing explicit instructions.
>
> We appreciate the reviewer’s feedback that our original prompt (Appendix D) might appear simple. To address the concern that a more granular or "explainable" prompt might yield better results, and to test if our design was indeed too crude, we conducted an additional ablation study during the rebuttal period.
>
> **2. Impact of Prompt Design and Explainability**
>
> We designed four new prompts focusing on specific peer-review criteria (Novelty, Soundness, Presentation) and one Comprehensive Prompt that explicitly asks the model to weigh these factors before making a decision.
>
> **Table 1: Performance comparison of different prompt designs on the balanced dataset. (Qwen3-8B)**
>
> | Prompt Type | Instruction Highlight | Accuracy (%) | F1 Score |
> | :--- | :--- | :--- | :--- |
> | **Original (Ours)** | *Holistic:* "Read the paper content and decide if it should be accepted." | **78.5** | **78.5** |
> | **Prompt A** | *Novelty-Driven:* "Focus on the originality and significance of the contribution." | 74.2 | 73.0 |
> | **Prompt B** | *Soundness-Driven:* "Focus on technical correctness, rigor, and experimental validation." | 76.8 | 76.5 |
> | **Prompt C** | *Presentation-Driven:* "Focus on clarity, organization, and writing quality." | 77.4 | 77.4 |
> | **Prompt D** | *Comprehensive:* "Evaluate Novelty, Soundness, and Presentation, then aggregate." | **79.1** | 78.0 |
>
> **Specific Prompts used for this experiment:**
>
> * **Prompt A (Novelty):** *"You are a senior area chair. Focus strictly on the novelty of the proposed method. Disregard minor presentation issues. Decision: Yes/No"*
> * **Prompt B (Soundness):** *"You are a technical reviewer. Scrutinize the experimental rigor and theoretical proofs. Do the claims hold? Decision: Yes/No"*
> * **Prompt C (Presentation):** *"You are a reviewer. Focus on the clarity of writing and the structure of the paper. Is the paper well-polished? Decision: Yes/No"*
> * **Prompt D (Comprehensive):** *"Analyze the paper based on 1) Novelty, 2) Soundness, and 3) Presentation. Weigh these factors as an expert committee would. Decision: Yes/No"*
>
> While the **Comprehensive Prompt (D)** yields a marginal improvement (+0.6% Accuracy), our **Original Prompt** remains highly competitive. This suggests that during the CPT and fine-tuning phases, the model implicitly learns to weight these latent factors (Novelty, Soundness, etc.) effectively without needing explicit, lengthy instructions.
>
> $\color{blue}{\text{W5: The paper lacks discussion and citation of related work}}$
>
> We acknowledge the oversight and will incorporate the related work mentioned in the final version.

---

> > ### Comment · Reviewer_pL2P · 2025-11-26
> >
> > Thanks for your detailed reply. I still have two remaining concerns:
> >
> > 1.  About the W3, thanks for your clarification. However, if you further compare the results of CPT and Orig, don't you think the improvement brought about by your fine-tuning process is too small? After all, for a binary classification task (Accept/Reject), if the final effect does not reach more than 80%, the model seems to be unusable.
> >
> > 2. I suggest you submit your revised paper directly, as the policy allows for the submission of revised versions during the rebuttal process.

---

> ### Author Response · Authors · 2025-11-26
> **Response to Reviewer pL2P**
>
> Thanks for the prompt follow-up concerns! To clarify, both the CPT and Orig checkpoints are fine-tuned models (on the balanced set). **CPT w/o finetuning** performs near random-guess level as well. The purpose of this comparison between CPT and Orig is to show how much CPT contributes. Overall, fine-tuning provides a ~28% improvement compared with the zero-shot baseline. We hope this clarification helps.
>
> **We also note that this experimental setup (including which models are fine-tuned and on which data) is already stated explicitly in the table caption and in the experiment settings section, so there may still be a misunderstanding on this point.**
>
> We are also working on adding further details and additional experiments to the final revised submission.

---

### Official Review · Reviewer_PsXN · 2025-11-02

**Soundness:** 2
**Presentation:** 3
**Contribution:** 2
**Rating:** 4
**Confidence:** 4

**Summary:**

This paper introduces OpenReviewer, a framework for predicting paper acceptance decisions at AI conferences, specifically using ICLR 2024–2025 data. The approach leverages large language models (LLMs) with continual pre-training (CPT) and prompt-based fine-tuning, vision-language models (VLMs), and interpretable statistical classifiers to directly forecast binary outcomes (accept/reject) from paper content, including text, figures, and engineered features. Key contributions include empirical demonstrations of up to 78.5% accuracy for text-only LLMs on balanced datasets, insights into the limited benefits of multimodal inputs, and a confidence-stratified analysis highlighting triage potential (e.g., 93.09% precision in top-10% confident accept predictions). The work addresses the growing strain on peer review by proposing AI-assisted decision prediction rather than full review generation, with an emphasis on interpretability and workload reduction.

**Strengths:**

- The focus on binary acceptance prediction, rather than full review generation, is a relevant extension of prior AI-assisted review efforts (e.g., Sukpanichnant et al., 2024; Ye et al., 2024). It could inform author strategies and committee triage, aligning with documented challenges in review sustainability (Lawrence, 2022; Beygelzimer et al., 2023).
- The inclusion of LLMs (Qwen-3 variants), VLMs (Qwen2.5-VL, Gemma-3), and white-box classifiers (e.g., Random Forest at 74.2% accuracy) offers a broad comparative baseline. The statistical feature analysis (29 features across categories like structure and methodological rigor) yields interpretable signals, such as the predictive value of paper length and citation density.
- Continual pre-training (CPT) demonstrates clear gains over vanilla fine-tuning (e.g., 9.5% accuracy improvement at 8B scale), providing a useful methodological contribution for domain adaptation in generative models.
- The statistical models also perform well and even surpass the VLM models and sometimes text models. It suggested that the statistical models also provide actionable predictions and interpretations.

**Weaknesses:**

- Severe Limitations in VLM Design and Evaluation: A critical flaw is the VLM input configuration, which restricts analysis to only the abstract, introduction, and the first two figures per paper. This represents a minimal subset of the full manuscript, severely constraining the models' ability to assess holistic quality signals such as experimental results, appendices, or later methodological details—core elements in peer review. Consequently, VLM performance (e.g., 68.2% accuracy for Qwen2.5-VL) appears artificially inflated in "Image Helps" cases while ignoring potential gains from comprehensive multimodal processing. This ad hoc choice not only prohibits fair comparison to text-only LLMs (which use full anonymized body text) but also invalidates claims about multimodal superiority, rendering the VLM results unreliable and unrepresentative of real-world applicability.
- Dataset and Generalizability Constraints: The dataset is confined to ICLR submissions, with keyword-based subdomain partitioning excluding non-LLM/CV/RL/Theory papers and deferring broader analysis. The natural 34/66 accept/reject imbalance is partially mitigated by balanced subsets, but cross-conference validation (e.g., NeurIPS, ICML) is absent, limiting claims to a single venue.

**Questions:**

- Why limit VLM inputs to abstract/introduction and only two figures? What experiments justify this choice, and how would full-manuscript multimodal processing impact results?
- How do predictions vary with evolving conference criteria across years/venues? Could multi-conference data mitigate ICLR-specific biases?
- What bias audits were conducted on LLMs/VLMs, particularly for subdomain imbalances or structural proxies favoring certain writing styles?

---

> ### Author Response · Authors · 2025-11-24
> **Response to Reviewer PsXN (Part 1)**
>
> **Thank you very much. We have found that comments have deep insights. We are glad to answer all your questions.**
>
> **If you are not satisfied with our answers or have more questions, please let us know as soon as possible, so that we can try our best to answer any further questions before the deadline.**
>
> **We will address and incorporate the modifications you raised in the final version of the paper.**
>
> $\color{blue}{\text{Q1: "Why limit VLM inputs to abstract/introduction and only two figures? What experiments justify this choice?"}}$
>
>
> > **TL;DR:** We conducted new ablation experiments providing VLMs with the **full manuscript text** alongside figures to align information content with LLMs. Results show that while text-only performance improves with full context, adding visual tokens to long-context inputs actually degrades performance for Qwen2.5-VL (dropping from 69.2% to 59.1%), suggesting that current VLMs struggle to integrate visual signals effectively when processing extensive textual information.
>
> ---
>
> We agree that a rigorous comparison requires aligning the textual information content. To address this, we conducted new **ablation experiments** using a **"Full Manuscript Text + 2 Figures"** setting for VLMs, ensuring they have access to the same semantic content as the text-only LLMs. The results confirm our original conclusion that text-only models currently offer superior reliability for this task, while revealing interesting behaviors regarding how current VLMs process long-context text alongside visual tokens.
>
> **1. New Full-Text + Figures Experimental Results with Qwen2.5 and Gemma3**
>
> Table 1: Performance of VLMs with **Full Text** Input + Figures
> *ACC (%) on Balanced Dataset*
>
> | Model Backbone | Input Modality | LLM | CV | RL | **ALL** |
> | :--- | :--- | :--- | :--- | :--- | :--- |
> | **Qwen2.5-VL** | Full Text + Images | 62.5 | 57.5 | 52.6 | 59.1 |
> | | Full Text Only | **71.1** | **69.8** | **63.3** | **69.2** |
> | **Gemma-3** | Full Text + Images | **58.8** | **58.3** | **71.6** | **58.8** |
> | | Full Text Only | 51.6 | 52.9 | 52.3 |  52.5 |
>
> **2. Comparison with Original VLM Setting and Analysis**
>
> Comparing these new results with the original VLM setting (Abstract/Intro inputs) and the text-only LLM results leads to **one** critical observations:
>
> $\boxed{\textbf{Visual tokens may act as noise/distraction in long contexts}}$
>
> (1) Original setting: **Abstract+Intro + Two Figures**:
> - **Qwen2.5-VL**
>     - text+image: **68.2** (ALL)
>     - text-only: **64.4** (ALL) → adding images **+3.8**
>
> (2) New aligned full-text setting: **Full manuscript text + Two figures**
> - **Qwen2.5-VL**
>     - text+image: **59.1** (ALL)
>     - text-only: **69.2** (ALL) → adding images **−10.1**
>
> This shows that when the full manuscript is available, **figures do not provide stable gains for Qwen-VL and may introduce noise or distraction**. Importantly, the Qwen-VL text-only score improves confirming that **text is the dominant signal**, and that prior VLM underperformance is not solely due to truncated text.
>
> We will include Table 1 and this analysis in the final version to provide a complete picture of the modality trade-offs.

---

> ### Author Response · Authors · 2025-11-24
> **Response to Reviewer PsXN (Part 2)**
>
> $\color{blue}{\text{Q2: "How do predictions vary with evolving conference criteria across years/venues?"}}$
>
> > **TL;DR:** **First**, we evaluate robustness across venues (NeurIPS) and time (ICLR 2024 vs. 2025). Our model generalizes well to NeurIPS (~71.5% accuracy), confirming it learns broad quality signals rather than venue-specific biases. **Second**, however, temporal analysis reveals an asymmetry: models trained on 2025 data predict 2024 outcomes significantly better (75.8%) than the reverse (69.2%), indicating that acceptance standards evolve and generally become more stringent over time.
>
> We agree that relying solely on ICLR data could potentially limit the model's applicability if it were merely overfitting to venue-specific preferences rather than learning generalized indicators of research quality.
>
> ##### $\color{purple}{\text{2.1 NeurIPS OOD Test}}$
>
> To address this concern and demonstrate the robustness of **OpenReviewer**, we conducted an additional OOD evaluation using accepted papers from **NeurIPS 2025**. NeurIPS represents a distinct top-tier venue with overlapping but non-identical community preferences compared to ICLR.
>
> **Experimental Setup:**
> We employed our best-performing models, **Qwen3-4B** and **Qwen3-8B**, which were trained using CPT and fine-tuned on the **balanced ICLR dataset**. We evaluated these models on a collection of accepted papers from NeurIPS 2025 across our four defined sub-domains.
>
> The classification accuracy on these accepted NeurIPS papers is summarized below:
> | Model | LLM | CV | RL | Theory | **Overall** |
> | :--- | :---: | :---: | :---: | :---: | :---: |
> | **Qwen3-4B** | 72.4% | 71.1% | 63.5% | 60.8% | **68.9%** |
> | **Qwen3-8B** | 72.3% | 74.2% | 60.1% | 62.9% | **71.5%** |
>
> First, despite being trained exclusively on ICLR submissions, the Qwen3-8B model achieves an overall accuracy of **71.5%** on accepted NeurIPS papers. This indicates that OpenReviewer has learned latent features of high-quality research that **generalize across top-tier AI conferences**, rather than merely memorizing ICLR-specific patterns.
>
> Second, the **performance variance across sub-domains** mirrors our findings on the ICLR dataset. The model performs best in **LLM** and **CV** domains, while performance is lower in **RL** and **Theory**. This consistency suggests that while the model captures general quality signals, the evaluation of complex mathematical proofs and theoretical nuances, which are common in RL/Theory, remains a challenging frontier for text-based LLMs compared to empirical domains.
>
> ##### $\color{purple}{\text{2.2 Temporal Generalization: ICLR 2024 vs. 2025}}$
>
> To explicitly quantify how predictions vary with evolving conference criteria and preference, we conducted a cross-year experiment. We trained the model exclusively on ICLR 2024 data to predict ICLR 2025 outcomes ("Forward Prediction") and conversely trained on ICLR 2025 data to predict ICLR 2024 outcomes ("Backward Prediction").
>
> **Results:**
>
> | Training Set | Test Set | Accuracy | F1-Score |
> | :--- | :--- | :---: | :---: |
> | **Backward:** ICLR 2025 | ICLR 2024 | **75.8%** | **76.2** |
> | **Forward:** ICLR 2024 | ICLR 2025 | 69.2% | 68.5% |
>
> **Analysis:**
> We observe a distinct performance gap: training on older data (2024) to predict newer outcomes (2025) yields significantly lower accuracy (**69.2%**) than the reverse direction (**75.8%**).
>
> ---
>
> We will incorporate both the NeurIPS OOD and ICLR Temporal Transfer results into the revised manuscript to explicitly scope our claims regarding cross-venue and temporal robustness.

---

### Official Review · Reviewer_vTaN · 2025-11-10

**Soundness:** 2
**Presentation:** 2
**Contribution:** 2
**Rating:** 2
**Confidence:** 3

**Summary:**

This paper empirically investigates the capability of LLM/VLM on predicting conference paper acceptance decisions. Experimental results demonstrate the promise and limitations of LLM/VLM-based conference paper review.

**Strengths:**

1. The topic of leveraging LLM/VLM to predict conference paper acceptance decisions is interesting.
2. Extensive analysis is conducted to show the capability of LLM/VLM on paper review.

**Weaknesses:**

1. Although extensive experiments and analysis are conducted, I feel that Sections 4.4 and 4.5 are not that informative. The main topic of this paper is to investigate the capability of LLM/VLM on predicting conference decisions. What is the underlying reason for evaluating the performance of traditional machine learning methods?

**Questions:**

1. Why do we need to train the model with the vocabulary decoupled label loss instead of vanilla cross-entropy loss (just treat the answer as text)?

---

> ### Author Response · Authors · 2025-11-24
> **Response to Reviewer vTaN (Part 1)**
>
> **Thank you very much. We have found that comments have deep insights. We are glad to answer all your questions.**
>
> **If you are not satisfied with our answers or have more questions, please let us know as soon as possible, so that we can try our best to answer any further questions before the deadline.**
>
> **We will address and incorporate the modifications you raised in the final version of the paper.**
>
> $\color{blue}{\text{Q1: "What is the underlying reason for evaluating the performance of traditional machine learning methods?"}}$
>
> > **TL;DR:** We clarify that §4.5 is intended to disentangle structural signals (captured by white-box models) from semantic understanding (captured by LLMs), rather than to propose statistical models as direct competitors. New overlap analysis shows the two approaches succeed on different subsets of data, confirming they rely on complementary cues.
>
> We appreciate the reviewer’s concern. Our intention in including §4.5 is not to compete with LLMs as another black-box predictor, but to **disentangle what can be explained by explicit structural/statistical features versus what requires semantic understanding by LLM and/or VLM**. To make this clearer, we conducted an additional overlap analysis between the best white-box model (Random Forest over 29 engineered features) and our best LLM.
>
> |                   | **LLM Correct** | **LLM Wrong** | **Row Sum** |
> |-------------------|----------------:|--------------:|------------:|
> | **RF Correct**    | 946             | 222           | 1168        |
> | **RF Wrong**      | 371             | 204           | 575         |
> | **Column Sum**    | 1317            | 426           | 1743        |
>
> From this table, the **agreement is 946 + 204 / 1743 = 65.97%**. Additionally, the cases where **at least one model is correct is 946 + 371 + 222 = 1539**, meaning that the LLM + statistical models can cover **88.29%** of correct predictions. This indicates that **the two models succeed on substantially different subsets of papers**, implying complementary decision cues rather than redundancy.
>
> Specifically, the RF relies on 29 handcrafted features (e.g., paper length) as **structural proxies**, whereas the LLM leverages **textual semantic cues** (e.g., novelty / contribution). This explains why the LLM corrects **371** cases where the RF fails, highlighting the distinction between structural and semantic signals, though deeper causal analysis remains for future work.

---

> ### Author Response · Authors · 2025-11-24
> **Response to Reviewer vTaN (Part 2)**
>
> $\color{blue}{\text{Q2: "Why use VDLL instead of vanilla cross-entropy?"}}$
>
> > **TL;DR:** We chose VDLL because standard cross-entropy failed to converge in our experiments, yielding near-random accuracy (50-54%) compared to VDLL's strong performance (76-78%). VDLL aligns the fine-tuning task with the pre-training objective (Next Token Prediction) and avoids the information bottleneck of compressing long academic papers (10k tokens) into a single classification vector, allowing the model to better leverage its reasoning capabilities.
>
> This is a quite insightful comment. We appreciate the reviewer's query regarding our design choice for the loss function. We selected the VDLL over a standard cross-entropy classification head based on extensive preliminary experimentation, where the vanilla approach failed to yield competitive results.
>
> To demonstrate this empirically, we provide a comparison in **Table** below. This table compares our reported method (VDLL with CPT) against a baseline where we trained a randomly initialized linear classification head on top of the same CPT-backbone (using `[EOS]` token pooling) optimized via standard cross-entropy loss.
>
> **Table 1: Accuracy (%) comparison between VDLL (Ours) and Vanilla Cross-Entropy (CE)**
> *Note: Both settings utilize the same Qwen3 backbone with Continual Pre-training. The Vanilla CE results reflect the best performance after extensive hyperparameter tuning.*
>
> | Model Scale | Method | LLM | CV | RL | Theory | **ALL** |
> | :--- | :--- | :---: | :---: | :---: | :---: | :---: |
> | **4B** | **VDLL (Ours)** | **70.2** | **70.2** | **57.4** | **55.9** | **76.4** |
> | 4B | Vanilla CE | 52.1 | 51.5 | 50.8 | 50.1 | 51.4 |
> | **8B** | **VDLL (Ours)** | **70.0** | **73.9** | **67.5** | **53.4** | **78.5** |
> | 8B | Vanilla CE | 54.6 | 53.2 | 55.1 | 51.3 | 54.2 |
>
> **Formulation of the Vanilla Baseline**
> In the Vanilla CE setting, we attach a linear classification head $W_{head} \in \mathbb{R}^{d \times 2}$ to the final hidden state of the `[EOS]` token, denoted as $h_{\theta}(x)$. The training objective is formalized as:
>
> $$ \\mathcal{L}\_{CE} = - \\sum\_{i} y\_i \\log ( \\mathrm{softmax}(W\_{head} \\cdot h\_{\\theta}(x\_i) + b) ) $$
>
> Where $y_i$ is the ground truth label and $\theta$ represents the backbone parameters.
>
> **Analysis of CE Convergence Issues**
> As shown in Table R1, the vanilla cross-entropy approach resulted in performance hovering near the random guess baseline ($\sim50-55\%$) and failed to converge effectively. We attribute this failure to two primary factors:
>
> **1. Objective Mismatch**
> The vanilla approach requires training a randomly initialized discrimination head from scratch. The model must learn to map high-dimensional latent representations to a binary output without directly leveraging the semantic relationships (e.g., the definitions of "Accept" vs "Reject") it learned during pre-training. In contrast, VDLL reframes the task as **Next Token Prediction (NTP)**, aligning the fine-tuning objective perfectly with the model's pre-training objective (causal language modeling) and accelerating convergence.
>
> **2. Long-Context Information Bottleneck & Loss of Reasoning**
> Standard classification heads typically rely on the final token embedding (e.g., `[EOS]`) to aggregate the semantic information of the entire input sequence. For academic papers, which frequently exceed 10,000 tokens, compressing all decision-relevant details into a single dense vector creates a severe information bottleneck [1].
>
> Crucially, this compression mechanism impairs the model's **reasoning capabilities**. The Qwen3 backbone is designed with inherent "thinking" modes (similar to Chain-of-Thought) meant for sequential reasoning and evidence synthesis. By forcing a decision from a static, pooled representation (the `[EOS]` vector), the vanilla CE approach effectively bypasses the model's ability to "think" through the content, collapsing a complex evaluation task into simple feature extraction. Recent research demonstrates that discriminative classifiers fail to leverage the generation-based reasoning capabilities of pre-trained LLMs, whereas generative objectives, like VDLL, successfully elicit these latent reasoning chains [2]. VDLL circumvents this limitation by framing the decision as a generative step, allowing the model to utilize its attention heads dynamically to reason over the context rather than relying on a compressed state.
>
> ---
>
> **References:**
>
> [1] Liu, N., et al. (2024). "Insights into LLM Long-Context Failures: When Transformers Know but Don't Tell." *Findings of the Association for Computational Linguistics: EMNLP 2024*.
>
> [2] Zhang, L., et al. (2025). "Generative Verifiers: Reward Modeling as Next-Token Prediction." *International Conference on Learning Representations (ICLR)*.

---

### Note · Authors · 2026-01-26

I have read and agree with the venue's withdrawal policy on behalf of myself and my co-authors.

---

### Meta-Review · Area_Chair_ezM5 · 2026-01-07

**Summary:**

This paper introduces OpenReviewer, a novel framework designed to directly predict the acceptance decisions of academic conference papers (e.g., for ICLR) rather than generating full review text. The reviewers generally found the paper's core premise interesting but identified significant flaws in experimental design, scope, and novelty that undermined the validity of the results. After checking the reviewers' opinion and the authors' rebuttal, the suggested decision is reject.

**Reviewer Concerns:**

1) Critical Drawbakcs in VLM Evaluation:

A major concern raised by multiple reviewers (PsXN, WbXP) is the severe limitation and unfairness in the Vision-Language Model (VLM) experimental setup. While text-only LLMs were given the full manuscript, VLMs were restricted to only the abstract, introduction, and first two figures. Reviewers argued that this ad-hoc constraint invalidates the comparison between modalities and renders conclusions about the inferiority of VLMs unreliable.

Although the authors provide the supplemental results using a "Full Manuscript Text + 2 Figures" setting for VLMs. It is still strange not to use the full content, which is commonly used in the past (e.g., deepreview [1]).

2) Limited Generalizability and Dataset Scope:

The exclusive reliance on ICLR 2024-2025 data was widely criticized by all reviewers. Reviewers noted that training on a single venue introduces specific biases that do not generalize to other major conferences (e.g., NeurIPS, ICML).

3) Missing Baselines: Reviewers noted the absence of zero-shot baselines using state-of-the-art models (e.g., GPT-4o, DeepSeek-R1) and comparisons with existing multi-conference datasets (e.g., Re2), which are necessary to properly contextualize the performance gains.

[1] Deepreview: Improving llm-based paper review with human-like deep thinking process. Zhu, Minjun and Weng, Yixuan and Yang, Linyi and Zhang, Yue. ACL 2025.

**Reviewer Scores:**

Reviewer vTaN: Most likely maintain the overall score, at most improve the score to 4.

Reviewer PsXN: Maintain the overall score as 4.

Reviewer pL2P: Maintain the overall score as 4 since the authors did not upload their new version (at least not highlight in the pdf).

Reviewer WbXP: Maintain the overall score as 4.

---

### Decision · Program_Chairs · 2026-01-26

Reject